# Rab27a co-ordinates actin-dependent transport by controlling organelle-associated motors and track assembly proteins

Noura Alzahofi[1,10], Tobias Welz[2,10], Christopher L. Robinson[1], Emma L. Page[1], Deborah A. Briggs[1], Amy K. Stainthorp [1], James Reekes[1], David A. Elbe[1], Felix Straub[2], Wouter W. Kallemeijn [3], Edward W. Tate [3], Philip S. Goff [4], Elena V. Sviderskaya [4], Marta Cantero[5,6], Lluis Montoliu [5,6], Francois Nedelec[7], Amanda K. Miles [8], Maryse Bailly [9], Eugen Kerkhoff[2] & Alistair N. Hume [1✉]

Cell biologists generally consider that microtubules and actin play complementary roles in long- and short-distance transport in animal cells. On the contrary, using melanosomes of melanocytes as a model, we recently discovered that the motor protein myosin-Va works with dynamic actin tracks to drive long-range organelle dispersion in opposition to micro-tubules. This suggests that in animals, as in yeast and plants, myosin/actin can drive long-range transport. Here, we show that the SPIRE-type actin nucleators (predominantly SPIRE1) are Rab27a effectors that co-operate with formin-1 to generate actin tracks required for myosin-Va-dependent transport in melanocytes. Thus, in addition to melanophilin/myosin-Va, Rab27a can recruit SPIREs to melanosomes, thereby integrating motor and track assembly activity at the organelle membrane. Based on this, we suggest a model in which organelles and force generators (motors and track assemblers) are linked, forming an organelle-based, cell-wide network that allows their collective activity to rapidly disperse the population of organelles long-distance throughout the cytoplasm.

[1] School of Life Sciences, University of Nottingham, Nottingham NG7 2UH, UK. [2] University Hospital Regensburg, Regensburg, Germany. [3] Department of Chemistry, Imperial College London, Molecular Sciences Research Hub, London W12 0BZ, UK. [4] Cell Biology and Genetics Research Centre, St. George's, University of London, London SW17 0RE, UK. [5] Centro Nacional de Biotecnologia (CNB-CSIC), Madrid 28049, Spain. [6] CIBERER-ISCIII, Madrid, Spain. [7] Sainsbury Laboratory, Cambridge University, Cambridge CB2 1LR, UK. [8] John van Geest Cancer Research Centre, Nottingham Trent University, Nottingham NG11 8NS, UK. [9] UCL Institute of Ophthalmology, 11-43 Bath St, London EC1V 9EL, UK. [10] These authors contributed equally: Noura Alzahofi, Tobias Welz. ✉email: Alistair.hume@nottingham.ac.uk

In animal cells, unlike plants and yeast, microtubules (MTs) and actin filaments (AFs) are thought to regulate transport in a manner akin to the infrastructure of a developed nation[1–4]. This 'highways and local roads' model suggests that MTs are tracks for long-range transport (highways) between the cell centre and periphery, driven by kinesin and dynein motors. Meanwhile AFs (local roads) and myosin motors work downstream picking up cargo at the periphery, and transporting it for the 'last μm' to its final destination. This model makes intuitive sense as MTs in many cultured animal cells form a polarised radial network of tracks spanning >10 μm from the centrally located centrosome to the periphery and appear ideally distributed for long-range transport. Meanwhile, with some exceptions in which AFs form uniformly polarised arrays, e.g., lamellipodia, filopodia and dendritic spines, AF architecture appears much more complex. In many cells AF appear to comprise populations of short (1–2 μm length), with random or antiparallel filament polarity, and not an obvious system of tracks for directed transport[5,6].

This view is exemplified by the co-operative capture (CC) model of melanosome transport in melanocytes[7,8]. Skin melanocytes make pigmented melanosomes and then distribute them, via dendrites, to adjacent keratinocytes, thus providing pigmentation and photo-protection (reviewed in ref. [9]). The CC model proposes that transport of melanosomes into dendrites occurs by sequential long-distance transport from the cell body into dendrites along MTs (propelled by kinesin/dynein motors), followed by AF/myosin-Va-dependent tethering in the dendrites. Consistent with this, in myosin-Va-null cells melanosomes move bidirectionally along MTs into dendrites, but do not accumulate therein, and instead cluster in the cell body[7,10]. This defect results in partial albinism in mammals due to uneven pigment transfer from melanocytes to keratinocytes (e.g., dilute mutant mouse and human Griscelli syndrome (GS) type I patients; Fig. 1a)[11,12]. Subsequent studies revealed similar defects in mutant mice (and human GS types II and III patients) lacking the small GTPase Rab27a (ashen) and its effector melanophilin (Mlph; leaden) which recruit and activate myosin-Va on melanosomes[8,13,14].

Previously, we tested the CC model by using cell normalisation technology to quantitatively examine the contribution of MTs and AF/myosin-Va to melanosome transport[15]. Surprisingly, our results indicated that MTs are essential for perinuclear clustering, but not peripheral dispersion of melanosomes. Instead we found that MTs retard dispersion, which is dependent upon myosin-Va and a population of dynamic AFs. Functional analysis of mutant proteins suggested that myosin-Va works as a processive motor dispersing melanosomes along AFs whose +/barbed ends are oriented away from melanosomes and towards the cell periphery. Finally, using an activatable motor to directly monitor melanosome dispersion in myosin-Va-null cells in real time, we found that myosin-Va can disperse melanosomes rapidly (~1 μm/min) into peripheral dendrites (>10 μm) even in MT-depleted cells. Overall our data highlighted the role of myosin-Va and dynamic AFs in long-range transport, rather than tethering, and suggest that melanosome distribution is determined by the balance of MT-dependent clustering and long-range AF/myosin-Va-dependent dispersion. However, studies of AFs organisation in melanocytes have not revealed the existence of a polarised network that would seem requisite for myosin-Va-driven transport and melanosome dispersion[7]. Thus, the mechanism of this process remained unclear.

Here, we investigated this issue and identify SPIRE1/2 and formin-1 (FMN1) AF assembly proteins as essential regulators of myosin-Va-driven melanosome dispersion. FMN1 is one of 15 mammalian formins that nucleate and elongate unbranched AFs (refs. [16,17]). FMN1 and FMN2 comprise the FMN subfamily of formins[18]. FMN1 function is linked to limb development, neurogenesis and spermatogenesis, while FMN2 acts in oocyte development, and memory and learning[19–25]. Like other formins, FMNs contains two formin homology domains (FH1 and FH2) that drive AF assembly[17]. Dimeric FH2 forms a ring that encircles the +/barbed ends of AFs, while the proline rich FH1 interacts with profilin in complex with G-actin promoting filament elongation[26–28]. The FMNs are characterised by a short (~30 amino acids) C-terminus FH2 tail that mediates the interaction with SPIRE proteins (termed FSI; formin/SPIRE interaction sequence)[29–31]. The N-terminal regions of the FMN1/2 formins are large (859 amino acids in murine FMN1), but contain no conserved sequence motifs.

Mammalian genomes encode two *SPIRE* genes, *SPIRE1* and *SPIRE2*. The SPIRE actin nucleators are modular proteins that contain an N-terminus AF nucleation module, comprising a KIND (kinase non-catalytic C-lobe domain) that interacts with the FSI motif of FMN1/2, and four G-actin-binding WH2 (WASP-homology 2) domains[29,32,33]. This is coupled to a C-terminus membrane-binding domain, comprised of SB (SPIRE box, conserved among SPIRE proteins) and FYVE-type zinc finger (Fab1p, YOTB, Vac1 and EEA1) domains[34,35]. The SPIRE box has sequences similarities to the N-terminal a-helix of Rab GTPase-binding domain (RBD) of the synaptic vesicle transport regulator Rabphilin-3A (RPH3A). Thus, FMN and SPIRE proteins may collaborate to assemble AFs at organelle membranes[22,23]. Previously, their combined function has been implicated in regulating oocytes development and repair of DNA damage[22,36,37]. In mouse oocytes, SPIRE1 and SPIRE2 cooperate with FMN2 to generate AFs for myosin-Vb-driven cortical transport of Rab[11] vesicles[22,23]. More recently work has identified a myosin-V globular tail domain binding motif (GTBM) located between the N- and C-terminus modules of SPIRE proteins, which may co-ordinate the recruitment of myosin-V and AF assembly to Rab[11]-positive intracellular membranes[38].

Here, we present evidence that the myosin-Va-mediated melanosome transport/dispersion in melanocytes is dependent upon AF assembly activities of FMN1 and SPIRE1/2, and that SPIRE1/2 (predominantly SPIRE1) can be recruited to melanosomes by Rab27a. Based on these findings, we propose a cargo-driven model of organelle dispersion in which Rab27a plays a central role co-ordinating the function of both AF motors and assembly proteins.

## Results

**FMN1, SPIRE1 and SPIRE2 are required for melanosome dispersion in melanocytes.** To better understand myosin-Va/dynamic AF-based melanosome dispersion in melanocytes, we used siRNA knockdown to test the involvement of known AF regulatory proteins in this process. For this we transfected wild-type melanocytes (melan-a) with an siRNA mini-library comprised of 130 pools (four target-specific oligonucleotide duplexes/pool) directed against the transcripts of known AF regulators (Supplementary Table 1). We then visually screened the transfected cells to identify siRNA that induced perinuclear melanosome clustering, reasoning that knockdown of proteins working with myosin-Va/dynamic AF should result in dispersion defects like those seen in myosin-Va/dynamic AF-deficient cells (Fig. 1a). Consistently (5/5 transfections), we found that knockdown of FMN1, and double knockdown of its interacting partners SPIRE1 and SPIRE2, induced melanosome clustering like that seen for Rab27a knockdown (part of a myosin-Va receptor at the melanosome), albeit that the extent of clustering was significantly lower (Fig. 1b, c; mean pigment area (% total); NT = 86 ± 10.28%, Rab27a = 31.28 ± 8.242%, FMN1 = 45.07 ± 8.061%

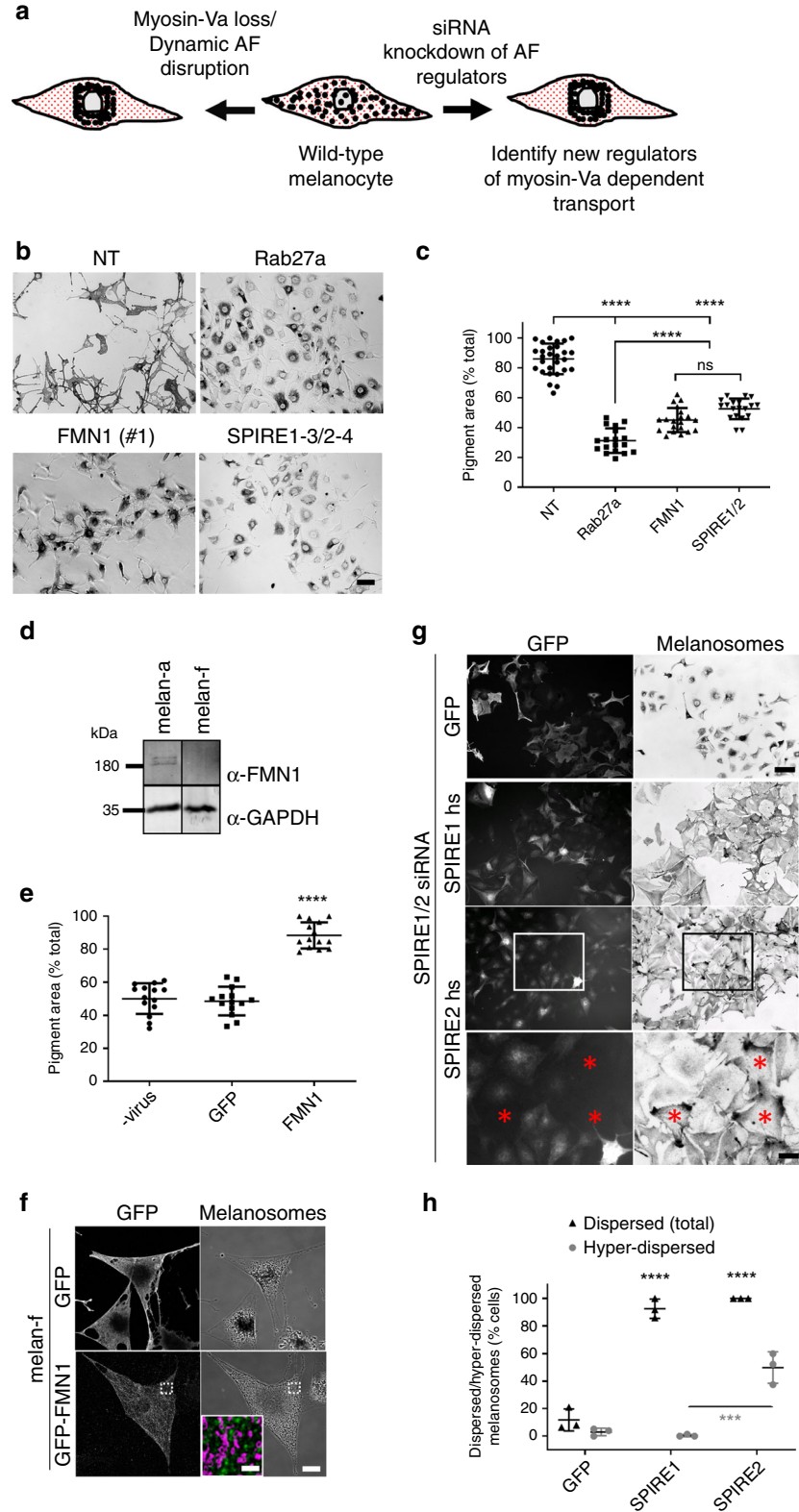

and SPIRE1/2 = 52.56 ± 6.868%). [Transfection with five of the other siRNA pools caused melanosomes clustering in melanocytes in >2/5 experiments. These targets were not investigated further.]

Interestingly, although quantitative real-time-PCR (Q-RT-PCR) confirmed that single siRNA transfection reduced mRNA for SPIRE1 and SPIRE2, only SPIRE1 knockdown resulted in a significant decrease in the proportion of cells with dispersed melanosomes compared with control (NT) siRNA-transfected cells (Supplementary Fig. 1a–e; dispersed melanosomes (% cells); NT = 86.43 ± 4.887% versus SPIRE1 = 46.93 ± 10.64% and SPIRE2 = 82.57 ± 13.26%). Nevertheless, the effect of SPIRE2 depletion could be seen by the significantly lower proportion of cells with dispersed melanosomes seen in SPIRE1/2 versus

**Fig. 1 FMN1- and SPIRE1/2-deficient melanocytes show perinuclear melanosome clustering. a** A schematic representation of the strategy used to identify regulators of myosin-Va/dynamic AF-dependent melanosome transport. In wild-type melanocytes melanosomes are dispersed throughout the cytoplasm. Loss of myosin-Va (or its regulatory proteins, e.g., Rab27a) or pharmacological disruption of dynamic AFs blocks melanosome dispersion, resulting in perinuclear melanosome clustering 15. To identify AF-associated proteins working with myosin-Va, we used siRNA to deplete known AF regulators and screened for targets whose depletion phenocopied the loss of myosin-Va/depletion of dynamic AFs, i.e., caused perinuclear clustering of melanosomes. **b** melan-a cells were transfected with the indicated siRNA fixed 72 h cells later and imaged using bright-field optics to observe melanosome distribution (see Experimental procedures). **c** A bee swarm plot showing the extent of melanosome dispersion in individual cells transfected with the indicated siRNA ($n$ (cells) = 28 (NT), 18 (Rab27a), 20 (FMN1) and 22 (SPIRE1/2)). Horizontal bars indicate the populations that are being compared. n.s. indicates no significant difference between the populations as determined by one-way ANOVA. All other comparisons yielded highly statistically significant differences $p$ = <0.0001. **d** A western blot showing the expression of FMN1 and GAPDH (loading control) in whole cell lysates of melan-a and melan-f melanocytes. **e** A bee swarm plot showing the extent of melanosome dispersion in adenovirus-infected melan-f cells expressing the indicated proteins ($n$ (cells) = 14 for all conditions). **** indicates a statistically significant difference ($p$ = <0.0001) between this population and the others. No other statistically significant differences were observed. **f** Confocal micrographs showing the distribution and effect of GFP-FMN1 expression on melanosome distribution in melan-f cells. White dotted boxes in images indicates the region shown in high-magnification overlay image (GFP-FMN1 in green and melanosomes in magenta). **g** melan-a cells were transfected with the indicated siRNA. After 72 h cells were infected with adenovirus expressing GFP or GFP-SPIRE1/2 (human), fixed 24 h later and processed for immunofluorescence. Cells were then imaged using bright-field and fluorescence optics to observe melanosome and GFP distribution. Asterisks indicate cells with hyper-dispersed melanosome distribution. **h** A bee swarm plot showing the percentage of SPIRE1/2-depleted/adenovirus-infected melan-a cells in low-magnification (10×) fields of view, in which melanosomes were dispersed and hyper-dispersed. Scale bars = 50 μm (**b**, **g**) and 21 μm (**g** magnified portion), 10 μm (**f**) and 4 μm (**f** magnified portion). **c, e, h** **** and *** indicates statistical significance of differences of $p$ = <0.0001 and $p$ = <0.001 as determined by one-way ANOVA. Significance indicators above datasets indicate differences compared with GFP control. The horizontal bar indicates the datasets that are being compared. No other significant differences were observed. Bars indicate the mean and 25th and 75th percentile of data. Source data for **c, d, e** and **h** are provided in the Supplementary Source data file.

SPIRE1 alone depletion (Supplementary Fig. 1b–e; mean % of cells with dispersed melanosomes; SPIRE1/2 = 12.73 ± 5.878%). In addition, in a subset of SPIRE1-depleted cells we saw that melanosomes were cleared from the cell centre and enriched at the periphery. This pattern (termed 'hyper-dispersed' (HD)) differed from the uniform dispersion pattern seen in control NT siRNA-transfected cells, SPIRE2 or SPIRE1/2-depleted cells (Supplementary Fig. 1d, e; mean HD melanosomes (% total cells) = 6.324 ± 2.645). These data indicate that SPIRE1 plays a dominant role in establishing the uniform (physiological) melanosome dispersion pattern seen in wild-type melanocytes. Consistent with this Q-RT-PCR showed that FMN1 and SPIRE1 mRNA expression levels were comparable and exceeded that of SPIRE2 by fivefold (Supplementary Fig. 1f; mRNA copies (×103)/50 ng total RNA; SPIRE1 = 17.33 ± 1.883, SPIRE2 = 3.468 ± 0.5726 and FMN1 = 27.2 ± 4.17). We were unable to detect FMN2 expression using Q-RT-PCR, indicating that FMN2 is unlikely to be expressed in melanocytes. This suggests that FMN1 and SPIREs (predominantly SPIRE1) cooperate to disperse melanosomes in melanocytes.

We next used add-back experiments, in which human SPIRE1 and SPIRE2 (siRNA resistant), and mouse FMN1 proteins were expressed in SPIRE1/2 depleted melan-a and immortal FMN1-deficient melanocytes (melan-f) to confirm that melanosome clustering in siRNA experiments was specific for depletion of the expected target (Fig. 1d–h; Fig. 1e, mean pigment area (% total), −protein expression = 50.15 ± 9.228, GFP = 48.66 ± 8.618, FMN1 = 88.43 ± 7.972; Fig. 1h, mean % of cells with dispersed melanosomes (total); SPIRE1/2 knockdown with GFP expression = 11.83 ± 8.02, SPIRE1 = 92.58 ± 7.072, SPIRE2 = 94.67 ± 11.36). Interestingly, we saw that in cells expressing lower levels of SPIRE2 melanosomes were hyper-, rather than uniformly, dispersed (Fig. 1h, Supplementary Fig. 1g; mean % of cells with HD melanosomes; GFP = 3.088 ± 2.748, SPIRE1 = 0.4695 ± 0.8132, SPIRE2 = 49.96 ± 11.44). This indicates that SPIRE2 is less efficient in uniformly dispersing melanosomes compared with SPIRE1.

**SPIRE1/2 and FMN1 generate dynamic AFs that are essential for melanosome dispersal.** As FMN and SPIRE cooperate in AF assembly, we hypothesised that they collaborate in melanocytes to assemble dynamic AFs used by myosin-Va to disperse melanosomes[18]. To test this, we used latrunculin-A to deplete dynamic AFs in SPIRE1/2 and control NT siRNA-transfected melan-a and melan-f cells, and then observed the effects on melanosome distribution and melanosome-associated AF content. We found that this treatment reduced melanosome dispersion and AF content in control melan-a cells (as before), but not further reduce these parameters in melan-f and SPIRE1/2-depleted melan-a cells[15] (Fig. 2a–d; mean pigment area (% total) ± SD in the absence and presence of latrunculin-A; melan-a = 68.26 ± 9.956 versus 51.56 ± 12.14; melan-f = 45.48 ± 11.9 versus 45.89 ± 13.29; NT-transfected melan-a = 85.18 ± 8.549 versus 43.94 ± 16.09; SPIRE1/2-depleted melan-a 36.9 ± 9.668 versus 35.28 ± 10.26; Fig. 2e, f; melanosome-associated AF content integrated density (AU) ± SD; melan-a = 413.7 ± 186.1 versus 167.6 ± 75.5; melan-f = 241.6 ± 112.3 versus 241.9 ± 93.26; NT-transfected melan-a = 767.3 ± 331.9 versus 304.8 ± 237.8; SPIRE1/2 siRNA-transfected melan-a = 304.7 ± 145.7 versus 213.2 ± 74.68). Conversely, re-expression of GFP-Fmn1, but not GFP, increased AF content in melan-f cells (Fig. 2g, h; integrated density (AU) ± SD; GFP-Fmn1 = 128.6 ± 71.79 versus GFP = 47.13 ± 21.52). These data indicate that FMN1 (and SPIRE1/2) are important factors for the assembly of dynamic AFs that support myosin-Va-dependent melanosome dispersion.

To further investigate this, we used high-resolution field emission scanning electron microscopy (FESEM), and replica transmission EM to analyse the cytoskeleton network surrounding the melanosomes in melan-a cells and variants. In line with the results of fluorescence microscopy, high-magnification FESEM images revealed that melanosomes in melan-a cells were surrounded by a meshwork of filaments compatible with AFs on the basis of their diameter (mean filament diameter ± SD = 8.6 ± 0.3 nm, $n$ = 87) and morphology (dense bundles and branched networks, Fig. 3a–c). Melanosomes are fully embedded in the filament network, with filaments visible above and below melanosomes (Fig. 3g, coloured yellow and orange, respectively), and in many cases appear as integral part of the network as multiple AFs can be seen originating from the surface of melanosomes, bridging adjacent organelles (Fig. 3g, coloured blue; Fig. 3k, arrows). In contrast in melan-f cells, melanosome-associated filaments were almost entirely absent and instead

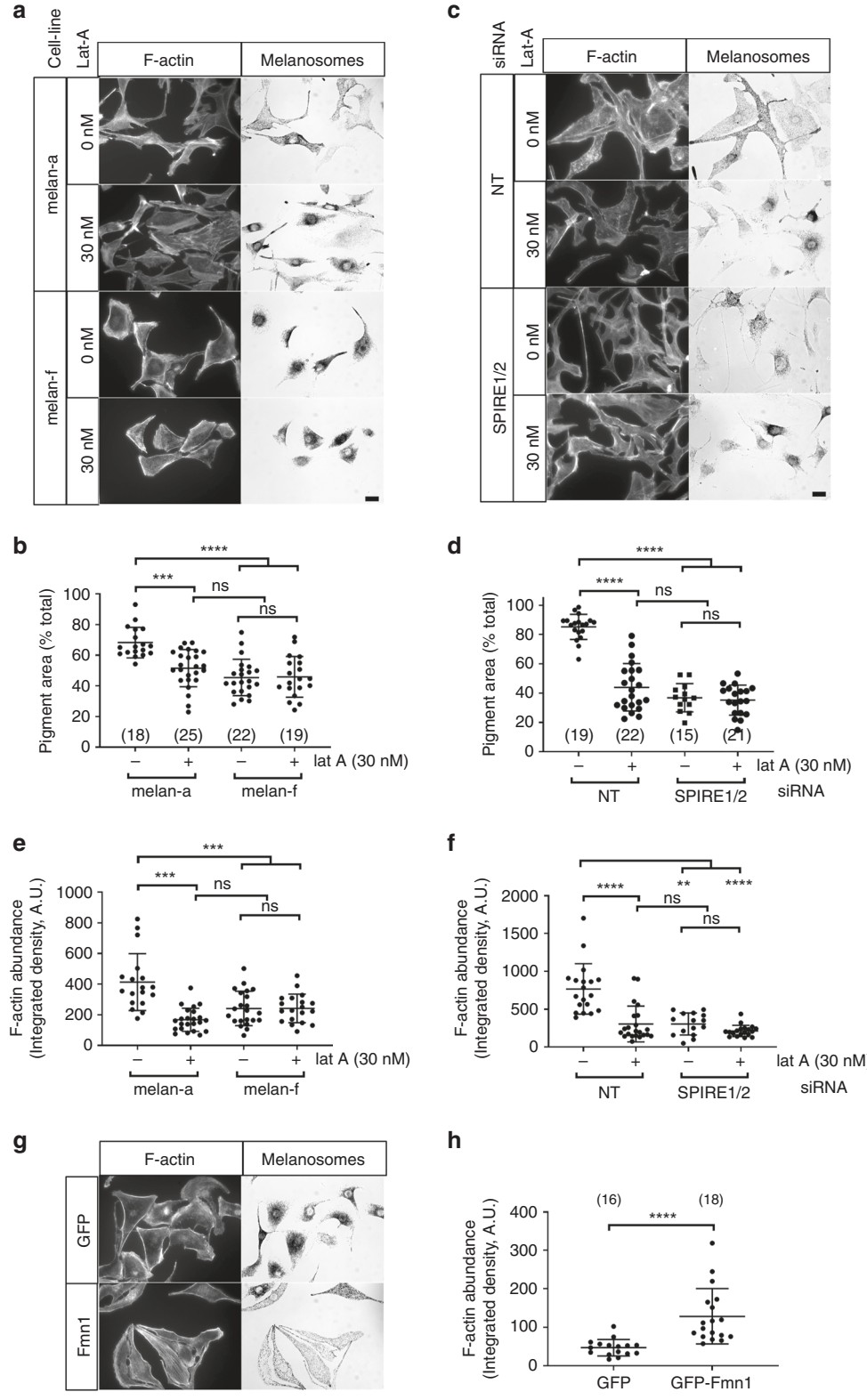

melanosomes were decorated with short filaments or 'stubs' (Fig. 3f, l, arrowheads). FESEM immuno-electron microscopy revealed that most of the filament network on, above and below the melanosomes was labelled by phalloidin, confirming that those filaments are part of a complex actin network (Fig. 3h, i). This was further confirmed by rapid freeze/freeze dry/replica TEM (ref. [39]), which revealed that myosin S1 decorated filaments surrounding melanosomes (Fig. 3j, arrowheads). To quantify the

differences in the melanosome-associated AFs, we further measured filament length of filaments emanating from melanosomes in FESEM images (Fig. 3k–m), confirming significantly shorter melanosome-associated filaments in melan-f cells. (Fig. 3m; mean filament length in nm ± SD: melan-a = 261.7 ± 160.6 (median = 223, $n = 433$) versus melan-f = 116.1 ± 72.32 (median = 103; $n = 443$); $p < 0.0001$. Similar differences in AF density and dimensions were observed in melan-a cells depleted

**Fig. 2 FMN1 and SPIRE1/2 generate latrunculin-A-sensitive AFs essential for melanosome dispersion.** melan-a and melan-f cells were plated onto glass coverslips, and transfected with siRNA as indicated: **c**, **d**, **f**, infected with GFP-FMN1 expressing virus **g**, **h** and/or incubated with latrunculin-A (lat-A) for 60 min **a**–**f** as indicated. Cells were then fixed and stained with fluorescent phalloidin to reveal AFs (see Experimental procedures). **a**, **c**, **g** Fluorescent and bright-field images showing the distribution of AFs and melanosomes in melanocytes. Scale bar = 15 μm. **b**, **d**, **e**, **f**, **h** Bee swarm plots showing the extent of melanosome dispersion (**b**, **d**) and AF abundance (**e**, **f**, **h**) in melanocytes, in the presence and absence of latrunculin-A. **b** and **e**, and **d** and **f** show data from the same population of cells. The number of cells measured in each case is indicated in brackets in the bee swarm plot associated with that data (**b**, **d**, **f**, **h**). Numbers of cells in **e** and **f** are the same as in **b** and **d**. **** and *** indicate significant difference $p = <0.0001$ and $p = <0.001$ as determined by one-way ANOVA. n.s. indicates no significant difference. Data are from one of three independent experiments. Bars within each dataset indicate the mean and 25th and 75th percentile of data. Bars linking datasets indicate the pairs that are being compared for similarity. Source data for **b**, **d**–**f**, **h** are provided in the supplementary Source data file.

of SPIRE1/2 and Rab27a (Supplementary Fig. 2). Given their dimensions, we suggest that these stubs correspond to short AFs that require FMN1 and SPIRE1/2 for extension into a network. These data support the hypothesis that SPIRE1/2 and FMN1 assemble AFs used by myosin-Va to disperse melanosomes and extend this idea by suggesting that these AFs are constructed at the melanosome membrane.

**The N-terminus AF nucleation (WH2) and FMN interaction (KIND) activities are essential for SPIRE function in melanosome transport.** To examine further their role in melanocytes, we tested which SPIRE protein domains are essential for melanosome dispersion. We found that neither the AF nucleation module (KW: FMN interaction (KIND) and G-actin binding (WH2 cluster)) or the membrane-binding module (MSFH: myosin-V-binding domain (M), SPIRE box (S), FYVE (F) and C-terminus flanking sequence (H)) fragments uniformly dispersed melanosomes as efficiently as intact SPIRE1 ($p = <0.001$) in SPIRE1/2-depleted melanocytes. However KW dispersed melanosomes to a significantly greater extent than either MSFH or GFP alone (Fig. 4a–c; mean % of cells with dispersed melanosomes, KW = $58.85 \pm 10.91$ ($p = <0.001$), MSFH = $18.69 \pm 4.37$ and GFP = $4.65 \pm 3.443\%$). Thus while both modules are essential for optimal SPIRE1 function, the AF nucleation module (KW) is the more significant element in driving melanosome dispersion. Interestingly, we noted that KW behaved similarly to SPIRE2 in promoting melanosome hyper-dispersion in a significant subset of cells (Fig. 4; mean % of cells with HD melanosomes KW = $36.69 \pm 7.746$; SPIRE2 = $35.91 \pm 10.72$; SPIRE1 = 0). This indicates that the SPIRE1 membrane targeting domain is essential for uniform cytoplasmic melanosome dispersion, and that SPIRE1 and SPIRE2 differ in their association with membranes.

To investigate AF assembly module further, we tested WMSFH and KMSFH truncations lacking either the KIND (FMN interaction) or WH2 (G-actin interaction) domains. We found that neither truncation dispersed melanosomes in SPIRE1/2-depleted cells to a greater extent than GFP (Fig. 4b, c; mean % of cells with dispersed (total) melanosomes; WMSFH = $19.01 \pm 1.693$ and KMSFH = $23.08 \pm 2.374$). This indicates that cooperation between SPIRE1 and FMN1 is required for AF assembly in melanocytes. Similar results were seen in experiments using human SPIRE2 to rescue SPIRE1/2 depletion (Supplementary Fig. 3). The integrity of the expressed SPIRE1 and SPIRE2 proteins (and FMN1 proteins below) was confirmed by western blotting (Supplementary Fig. 4a–c).

**AF assembly and SPIRE interaction domains are required for FMN1 function in melanosome transport.** We then used a similar approach to examine the role of FMN1 in melanocytes. We found that expression of a C-terminus fragment, encompassing the conserved FH1 and FH2 domains, and the FSI motif (FH1-FH2-FSI), restored peripheral melanosome distribution in

melan-f cells to a similar extent as intact FMN1, but a reciprocal N-terminus fragment did not (Fig. 5; mean pigment area (% total); GFP = $50.69 \pm 14.29\%$, FMN1 = $89.92 \pm 6.598\%$, N-term = $51.44 \pm 10.17\%$, FH1-FH2-FSI = $89.28 \pm 8.512\%$). Truncations that removed the FSI or FH1 were unable to fully disperse melanosomes (Fig. 5; mean pigment area (% total); $\Delta$FSI = $79.52 \pm 12.34\%$, FH2-FSI = $42.77 \pm 13.89\%$). Also we found that point mutations predicted to disrupt either the FH2 contact with AF +/barbed ends (I1074A and K1229D) or the electrostatic FSI/SPIRE-KIND interaction (K1418E) significantly reduced FMN1 function in transport (Fig. 5; mean pigment area (% total), I1074A = $57.77 \pm 12.73\%$, K1229D = $77.75 \pm 12.53$ and K1418E = $72.1 \pm 17.34$ %)[26,31]. These results show that the FH1 and FH2 domains, and the FSI motif are essential for FMN1 function and indicate that FMN1 assembles AFs in collaboration with SPIRE in melanocytes.

**The membrane-binding module of SPIRE is related to the Rab-binding domain of Rab27/3 effectors.** Within the mammalian Rab family (>60 genes), proteins of the Rab27/3/8 branch regulate the transport of exocytic vesicles towards the plasma membrane[40]. The Slp/Slac (synaptotagmin-like protein/Slp lacking C2 domain) class of effectors of these GTPases have a common Rab binding domain consisting of a FYVE-type zinc finger flanked by α-helical regions (termed H1 and H2 hereafter) that make direct and essential with contacts Rab3/27 (refs. [41,42]). Previous studies reported sequence similarity between the SPIRE box (SB) of SPIRE proteins and the H1 helix of the of the Rab27/3 effector RPH3A (ref. [34]). These observations together with evidence of Rab3a:SPIRE1 interaction in vitro and the presence of an adjacent FYVE-type zinc finger in the membrane-binding region suggest that SPIREs could be Rab27 effectors (Fig. 4a)[43]. Consistent with this our sequence alignments showed a high similarity between the SB and the H1 regions of several Rab27/3 effectors, including Mlph and conservation of residues important in Rab27/3 interaction among this group (Supplementary Fig. 5a coloured asterisks). Phylogenetic analysis of the sequences of the putative RBDs of SPIRE proteins with those of other containing Rab27/3/8 effectors grouped SPIRE proteins into this family (Supplementary Fig. 5b). These observations support the idea that SPIRE protein function in melanosome transport could be regulated by Rab27a.

**GTP-dependent interaction of SPIRE proteins with active Rab27a.** To test this possibility, we performed GST pull-down experiments using bacterially expressed, purified GST-Rab27a fusion proteins (GTP-locked Rab27a-Q78L and GDP-locked Rab27a-T23N mutant proteins) and lysates of HEK293 cells transiently expressing Myc-epitope-tagged SPIRE1 and SPIRE2 (Fig. 6a–c). Western blotting revealed that GST-Rab27a-Q78L pulled-down greater quantities of Myc-tagged SPIRE1 and SPIRE2 compared with GST-Rab27a-T23N, and that SPIRE1 interacted more strongly with Rab27a-Q78L compared with

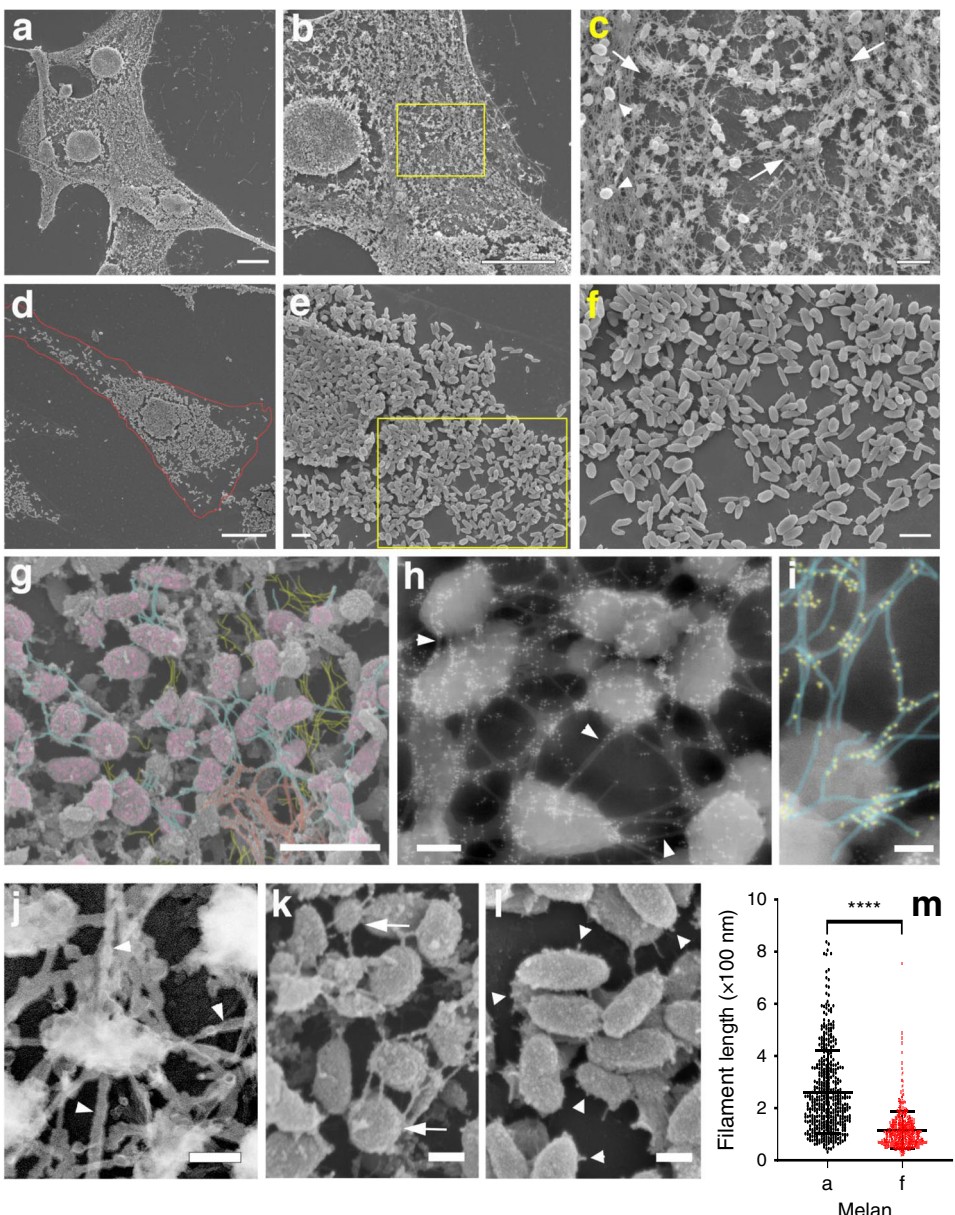

**Fig. 3 High-resolution electron microscopy reveals a reduction in melanosome-associated AFs in FMN1-deficient melanocytes compared with controls.**
**a**–**f** Wild-type (melan-a, **a**–**c**) and FMN1-deficient (melan-f, **b**–**f**) cells were prepared for field emission scanning electron microscopy (FESEM; see Experimental procedures). Cells in **a** and **d** are shown with increased magnification, with high magnification of the insets (yellow) in **c** and **f**. The red line in **d** indicates the cell outline. Arrows in **c** point at a loose network of filaments around melanosomes; arrowheads show melanosomes on top of a dense filament network. **g** Colourised filaments in high-magnification FESEM images of melan-a melanosomes (pink) indicate filaments linking (blue), above (orange) and below (yellow) melanosomes. **h**, **i** Immuno-electron microscopy of melan-a cells viewed with FESEM/backscatter showing phalloidin labelling (10 nm gold particles) of actin filaments around and over melanosomes, as well as inter-melanosome filaments (arrowheads). Higher magnification in **i** shows colourised filaments (cyan) and gold particle labelling (yellow). **j** TEM of rapid freeze/freeze dry metal replica showing myosin S1 decoration of filaments (arrowheads) around melanosomes. **g**, **k** High-magnification FESEM showing melanosomes with multiple AFs emerging from them in melan-a cells (**g**, arrows) or AF stubs on melan-f melanosomes (**l**, arrowheads). **m** Bee swarm plot showing size distribution for AFs emanating from melanosomes, as measured on FESEM images. *n* (actin filaments) = 433 (melan-a) and 443 (melan-f). **** indicates significant difference $p = <0.0001$ between melan-a and melan-f cells as determined by Mann–Whitney test. Source data for **i** are provided in the supplementary Source data file. Bars indicate the mean and 25th and 75th percentile of data. Scale bars: **a**, **b**, **d**, 10 μm; **c**, **e**–**g**, 1 μm; **h**, **k**, **l**, 200 nm; **i**, **j**, 100 nm.

SPIRE2. Similar results were obtained in experiments using wild-type GST-Rab27a loaded with GTPγS compared with GDPγS to pull-down GFP-SPIRE1-MSFH (Supplementary Fig. 6).

We then investigated Rab27a:SPIRE1/2 interaction in mammalian cells using a bimolecular fluorescence complementation (BiFC) assay, in which Rab27a and SPIRE1/2 were transiently co-expressed in HEK293 cells as fusions to C- and N-terminus fragments of the Venus yellow fluorescent protein (vYFP). We observed significantly higher BiFC signal in cells transfected with wild-type Rab27a and the Q78L mutant compared with the inactive mutants T23N and N133I, in which the mean BiFC signal was similar to cells transfected with the vYFP fragments alone (Fig. 6d; mean normalised BiFC (% max); SPIRE1/SPIRE2 WT = 89.35/85.41%, T23N = 28.35/5.969%, Q78L = 75.21/85.23%,

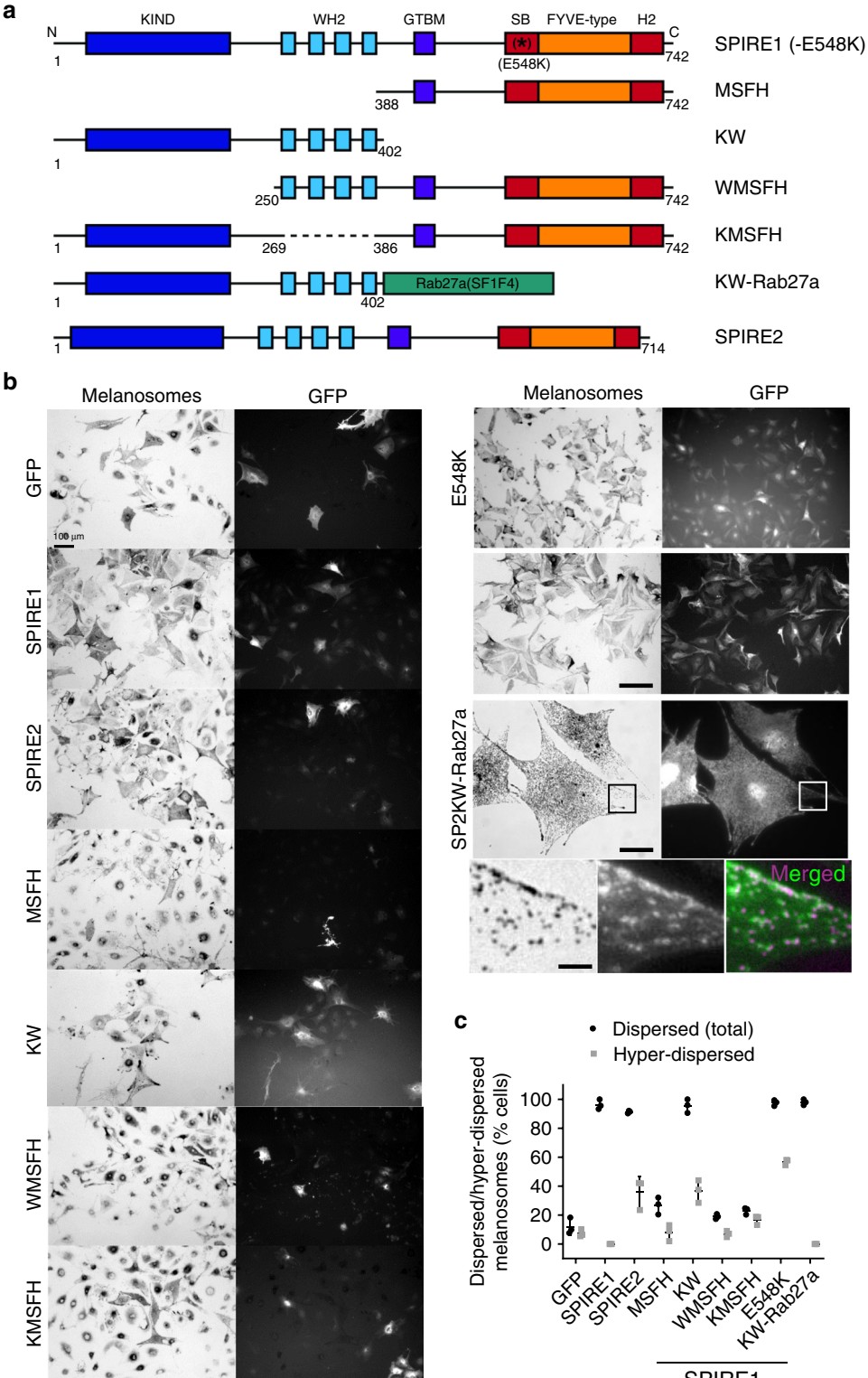

**Fig. 4 The FMN interaction (KIND) and AF nucleation (WH2) domains of SPIRE1 are essential for melanosome dispersion. a** A schematic representation of the domain structure of human SPIRE1, and the correspondence with mutant and chimeric proteins used in functional studies (**b**, **c**). Numbers indicate amino acid boundaries. K KIND, W WH2, M GTBM globular tail domain binding motif, S SB SPIRE box, F FYVE-type zinc finger, H C-terminal flanking sequences similar to H2 of Slp/Slac-proteins. **b** melan-a cells were depleted of SPIRE1/2 by siRNA transfection and 72 h later infected with adenoviruses expressing the indicated proteins. Cells were fixed 24 h later, processed for immunofluorescence and imaged using bright-field and fluorescence optics to observe melanosome and protein distribution/expression (see Experimental procedures). Scale bars = 100, 20 and 3 μm in low, medium and high-magnification images. Boxes in KW-Rab27a images indicate the region shown below. For the merged image green = KW-Rab27a and magenta = melanosomes. **c** Is a bee swarm plot showing the percentage of human SPIRE1/2 expressing SPIRE1/2-depleted melan-a cells (50 cells for each condition in each experiment), in which melanosomes are classed as dispersed and/or hyper-dispersed. Results shown are from of three independent experiments. Source data for **c** are provided in the Supplementary Source data file.

N133I = 24.71/6.601% and vYC alone = 8.637/16.65%). To examine Rab27a:SPIRE1/2 interaction in pigment cells, we expressed GFP-SPIRE1 (human) in B16-F1 cells, immuno-precipitated endogenous Rab27a from lysates using Rab27a specific antibodies and tested for co-immunoprecipitation of GFP-SPIRE1. Using mass spectrometry, we identified Rab27a (13 unique peptides), confirming the efficiency of the IP, and SPIRE1 (18 unique peptides) confirming the interaction of Rab27a and SPIRE1 in pigment cells (Supplementary Fig. 7, Supplementary Tables 2 and 3). These data concur with the results of pull-down assays and other approaches used here to test this interaction (Figs. 7 and 8, Supplementary Figs. 5–8) and further indicate that SPIRE1/2 are Rab27a effectors. Consistent with this recent proximity proteomic studies identified an interaction between Rab27a and endogenous SPIRE1 in HUVEC endothelial cells[44].

**The C-terminus membrane-binding module of SPIRE proteins interacts with Rab27a**. To map the Rab27a-binding site(s) in SPIRE proteins, we tested the ability of SPIRE1 truncations to interact with active Rab27a-Q78L using the GST- pull-down assay. Consistent with the distribution of sequence similarity between SPIREs and other Rab27 effectors, we found an interaction of GST-Rab27a-Q78L with Myc-SPIRE1-MSFH (and Myc-SPIRE2-MSFH) proteins but not Myc-SPIRE1-KWM (Fig. 6b). To better map the Rab27a-binding site within the SPIRE-MSFH, we generated further truncations and tested the Rab27a-Q78L interaction (Fig. 6a). We found that SPIRE1-SFH interacted strongly with Rab27a-Q78L as did the FYVE-only and SF proteins, albeit to a lesser extent (Fig. 6e). This indicates that the C-terminus SFH fragment of SPIRE proteins interacts with Rab27a, and suggests that the interaction mechanism is conserved with other Rab27 effectors.

**SPIRE proteins interact with Rab27a with lower affinity than Mlph**. To characterise Rab27a:SPIRE interaction further, we compared the affinity of this interaction with other Rab27a: effector interactions, e.g., Mlph. To do this, we quantified the affinity of interaction of GFP-SPIRE-MSFH and GFP-Mlph-RBD with GST-Rab27a-Q78L in pull-down assays by measuring GFP depletion from HEK293 cell lysates (Fig. 6a, f). This revealed dissociation constants ($K_d$) of 143 (±25) nM and 707 (±155) nM, and maximum binding levels of 39.4% and 12.0% for the GFP-Mlph-RBD and GFP-SPIRE1-MSFH. These data correspond to the previously determined $K_d$ of 112 nM Rab27a-Q78L:Mlph RBD interaction and indicate that SPIRE1 is a weaker Rab27a interactor than Mlph[45]. By this method, we were unable to determine the $K_d$ of SPIRE2:Rab27a interaction. These data align with pull-down assays results and suggest that Rab27a interacts more strongly with SPIRE1 than SPIRE2 (Fig. 6b).

**SPIRE1/2 associate with melanosomes by a Rab27a-dependent mechanism**. The above findings indicate that SPIRE proteins are Rab27 effectors whose expression is required for melanosome dispersion. As Rab27a is present on the cytoplasmic face of the melanosome membrane, this suggests that SPIRE proteins associate with melanosomes in a Rab27a-dependent manner. To test this, we expressed GFP-tagged SPIRE proteins and Rab27a in melan-a cells, and used confocal microscopy to examine their intracellular localisation. We observed that all three proteins were distributed in a punctate pattern throughout the cytoplasm (Fig. 7a–c). Consistent with Rab27a:SPIRE interaction studies, high-magnification imaging and intensity profile plots revealed that spots of SPIRE1 and Rab27a, but not SPIRE2, often overlapped with melanosomes (visible in phase contrast images; Fig. 7a–c; Pearson linear correlation coefficient = 0.823 for Rab27a, 0.623 for SPIRE1 and −0.081 for SPIRE2). We also saw that Myc-SPIRE1-MSFH overlapped

with melanosome resident protein tyrosinase-related protein 1 (Trp1) and other melanosome-targeted proteins (co-expressed mRuby3-Rab27a and GFP-myosin-Va-CC-GTD (that includes the melanocyte-specific coiled coil and globular tail domains; Supplementary Fig. 8)). These observations indicate that SPIRE1 associates with melanosomes.

To test whether this association was Rab27a dependent, we repeated the above experiment using melanocytes deficient in Mlph (melan-ln, in which Rab27a is retained on the melanosome membrane) and Rab27a (melan-ash)[46,47]. This revealed that Rab27a associated with melanosomes in both cell types (and dispersed melanosomes in melan-ash cells), but that SPIRE1 associated with perinuclear clustered melanosomes in melan-ln cells only (Fig. 7d, e; Pearson linear correlation coefficient = 0.601 and 0.739 for Rab27a, and = −0.230 and 0.617 for SPIRE1 in melan-ash and melan-ln). This indicates that association of SPIRE1 with melanosomes is dependent upon interaction with endogenous Rab27a, and that SPIRE1 is a Rab27 effector.

To further investigate Rab27a:SPIRE2 interaction in melanocytes, we developed a, sensitive, functional assay that measures changes in melanosome distribution in melan-ln melanocytes as a read-out of the interaction of candidate proteins with endogenous melanosome-associated Rab27a (Fig. 8a). We modified the previously described 'minimyosin' protein (that couples an active motor-lever arm (S1) fragment of myosin-Va to the RBD of the Rab27 effector synaptotagmin-like protein 2-a (Slp2-a)) by replacing the Slp2-a RBD fragment with SPIRE proteins to create 'myoSPIRE' fusions (Fig. 8a). We then tested the ability of these to disperse clustered melanosomes in melan-ln and melan-ash cells (i.e., in the presence/absence of endogenous Rab27a). In melan-ln, we found that both myoSPIRE proteins, like minimyosin but not Rab27a, dispersed melanosomes compared with GFP alone (Fig. 8b, c; mean pigment area (% total); GFP = 26.3 ± 7.43%, myoSPIRE1 = 63.4 ± 7.96%, myoSPIRE2 = 66.6 ± 3.76%, mini-Va = 62.3 ± 10.8 % for and Rab27a = 31.3 ± 10.7%). Interestingly SPIRE1 did not rescue melanosome transport, even though it can interact with Rab27a and myosin-Va like Mlph (Supplementary Fig. 9; mean pigment area (% total); GFP = 24.8 ± 7.91, SPIRE1 = 18.5 ± 3.95 and Mlph = 75.4 ± 18.8). One possibility is that the affinity of SPIRE1:myosin-Va/Rab27a interactions may be too low to recruit sufficient active myosin-Va to melanosomes to drive their dispersal. Consistent with this our data reveal the low affinity of Rab27a:SPIRE1/2 interaction relative to Rab27a:Mlph (Fig. 6f). Meanwhile previous studies showed that SPIRE2 interacts with myosin-Va-GTD (0.9 ± 0.11 μM) with lower affinity compared with Mlph (0.5 μM)[38,48]. In melan-ash, although Rab27a expression dispersed melanosomes, neither minimyosin nor myoSPIRE proteins did so to a significantly greater extent than GFP alone (Fig. 8c; mean pigment area (% total); GFP = 31.9 ± 6.80%, myoSPIRE1 = 37.8 ± 5.84%, myoSPIRE2 = 38.6 ± 5.97%, mini-Va = 39.2 ± 1.97% and Rab27a = 66.9 ± 6.20%). This shows that SPIRE1 and SPIRE2 can interact with endogenous Rab27a at the melanosome membrane in melanocytes. Consistent with this using confocal microscopy we observed that spots of both myoSPIREs were often located adjacent to melanosomes in melan-ln cells (arrows in Fig. 8b).

**Rab27 dependent targeting of SPIRE proteins to membrane is important for uniform melanosome dispersion in melanocytes**. Next, we used two approaches to test the functional importance of SPIRE:Rab27a interaction. Firstly, we exploited sequence conservation between SPIREs and other effectors to generated mutants that reduce SPIRE1:Rab27a interaction[41,49]. Using the GST Rab27a-Q78L pull-down assay, we found that mutation of

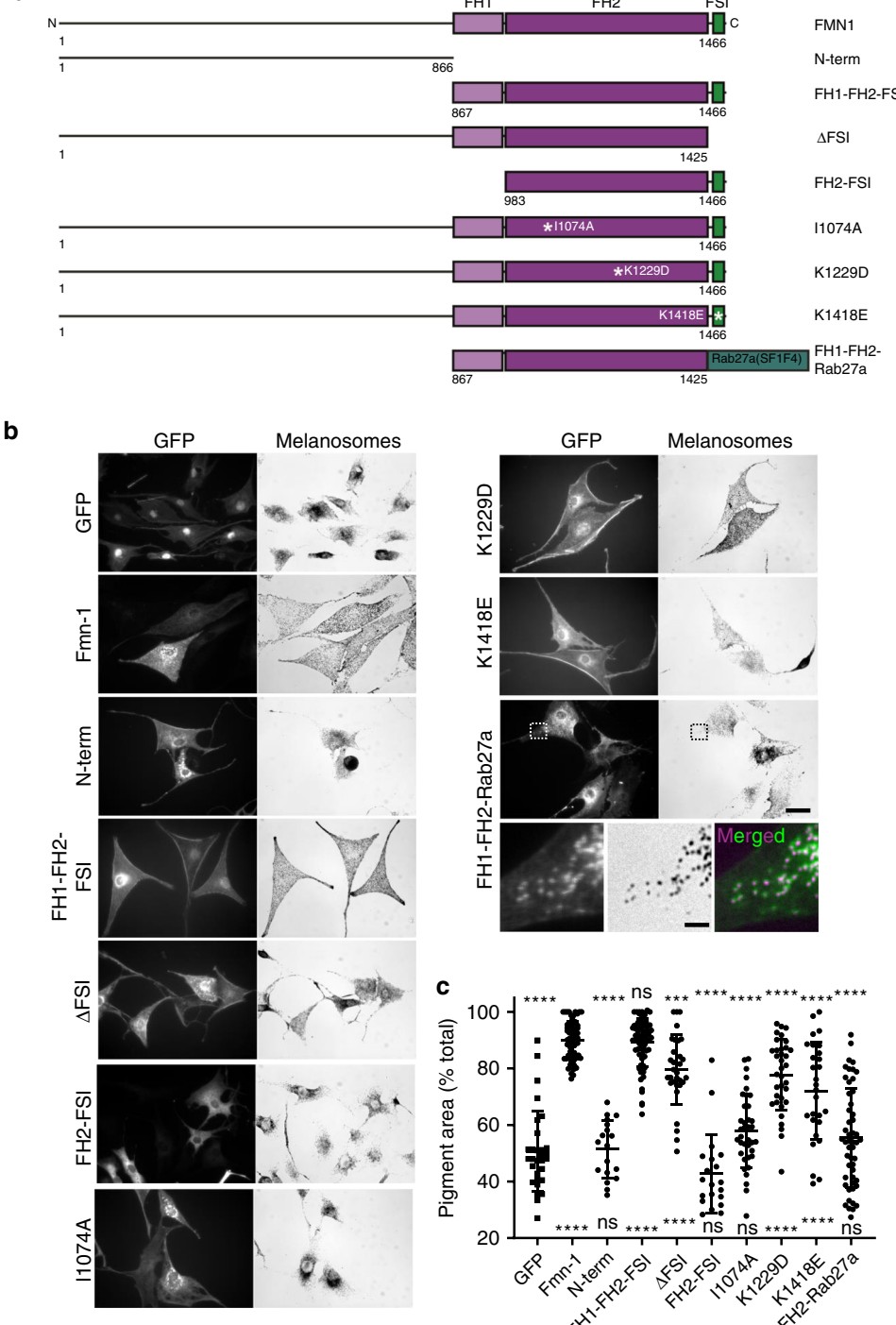

**Fig. 5 The AF assembly (FH1-FH2) and SPIRE interaction (FSI) domains of FMN1 are essential for melanosome dispersion. a** A schematic representation of the domain structure of murine FMN1 and the composition of truncation mutants, and chimeric proteins used in functional studies (**b**, **c**). White asterisks indicates the site of point mutations. Numbers indicate amino acid boundaries. **b** melan-f cells were plated on glass coverslips and infected with adenoviruses expressing the indicated proteins. Cells were fixed 24 h later, processed for immunofluorescence and the intracellular distribution of expressed protein and melanosomes (bright-field) was observed using a fluorescence microscope (see Experimental procedures). Scale bars = 20 μm and 2 μm for magnified region. Dashed boxes in FH1-FH2-Rab27a images indicate the region of the image shown in high magnification below. The merged image show melanosomes and FH1-FH2-Rab27a coloured magenta and green. **c** A bee swarm plot showing the extent of pigment dispersion in cells expressing the indicated proteins. $n = 30$ (GFP), 59 (FMN1), 18 (N-term), 70 (FH1-FH2-FSI), 31 (ΔFSI), 20 (FH2-FSI), 37 (I1074A), 35 (K1229D), 28 (K1418E) and 51 (FH1-FH2-Rab27a). ****, **, * and n.s. indicate significant differences of $p = <0.0001$, 0.01, 0.05 and no significant difference as determined by one-way ANOVA. Significance indicators above and below each dataset indicate differences between that dataset and the positive (FMN1 wild type) and negative (GFP alone) controls. Results shown are representative of three independent experiments. Bars indicate the mean and 25th and 75th percentile of data. FH formin homology, FSI formin-SPIRE interaction sequence. Source data for **c** are provided in the Supplementary Source data file.

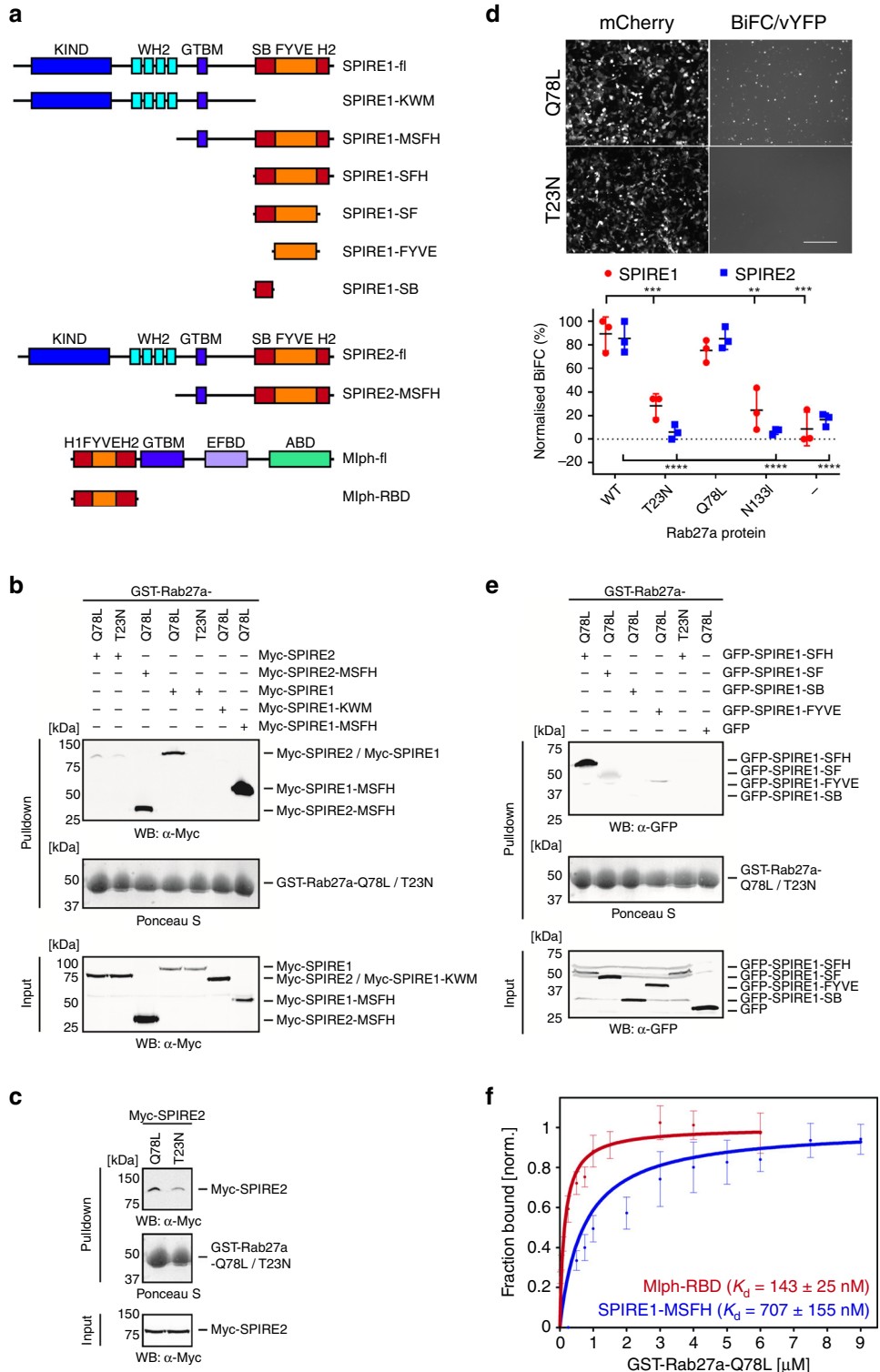

the highly conserved glutamic acid residue at codon 548 of human SPIRE1 to alanine or lysine (E548A/K) blocked interaction with Rab27a (Supplementary Fig. 5c). This is consistent with the possible electrostatic contact of the equivalent residue (E14/54) in Mlph/RPH3A and the basic side chain of arginine 80/83 located in the switch II region of Rab27b/Rab3a (Supplementary Fig. 5d)[41,42]. Expression of SPIRE1[E548K] in SPIRE1/2 depleted melan-a cells resulted in a significant increase in the proportion of cells with dispersed melanosomes compared with GFP (Fig. 4b, c; mean % of cells with dispersed melanosomes = 97.61 ± 2.044).

However, as seen for SPIRE1-KW and SPIRE2, in a significant proportion of cells expressing SPIRE1[E548K] melanosomes were hyper- rather than uniformly dispersed, as seen for intact SPIRE1 (Fig. 4b, c; mean % of cells with HD melanosomes = 56.85 ± 1.985). Also spots of SPIRE1[E548K] protein, like SPIRE2, were less frequently associated with melanosomes compared with wild-type SPIRE1 (Supplementary Fig. 5e arrows). Secondly, we investigated the functional effect of tethering the active SPIRE2 (KW) fragment to melanosomes, by expressing it as a fusion to the melanosome-targeted, non-functional, Rab27a[SF1F4] mutant

**Fig. 6 SPIRE1/2 interact with active Rab27a via their membrane-binding C-termini. a** A schematic representation of the domain structure of SPIRE1/2, Mlph and truncations used in interaction studies (**b**, **c**, **e**, **f**). Interaction of SPIRE1/2 and Rab27a was investigated using GST pull-down (**b**, **c**, **e**, **f**) and BiFC (**d**) assays (see Experimental procedures). **b**, **c**, **e** Western blots and Ponceau S stained filters showing the results of pull-down assays measuring the interaction of GST-Rab27a (active Q78L and inactive T23N mutants) with SPIRE1/2 and the indicated truncations (Myc-tagged (**b**, **c**) and GFP-tagged (**e**) in vitro). **c** is a contrast enhanced version of the section of **b** showing interaction of SPIRE2 with Rab27a-Q78L and Rab27a-T23N. **d** Fluorescence images and a bee swarm plot (upper and lower panels) showing the results of the BiFC assay, reporting the interaction of Rab27a with SPIRE1 and SPIRE2 in HEK293a cells (see Experimental procedures). Images of mCherry indicate transfection efficiency and vYFP indicates BiFC, i.e., interaction. The bee swarm plot shows the BiFC signal for populations of cells expressing SPIRE1/2 with and without active and inactive Rab27a mutants. Data shown are from three independent experiments. ****, *** and ** indicate significant differences of $p = <0.0001$, $p = <0.001$ and $p = <0.01$ between the adjacent dataset and Rab27a wild-type/SPIRE1/2 expressing cells as determined by one-way ANOVA of data for SPIRE1 and SPIRE2 with different Rab27a proteins. No other significant differences were observed. Two-way ANOVA comparison of BiFC signal for the Rab27a proteins with different SPIREs revealed no significant differences. Bars indicate the mean and 25th and 75th percentile of data. Scale bar = 250 μm. **f** Line plots showing the extent of binding of GFP-SPIRE1-MSFH ($n = 4$) and GFP-Mlph-RBD ($n = 3$) as a function of increasing GST-Rab27a-Q78L concentrations. Data are presented as mean values ± SEM and the equilibrium dissociation constants ($K_d$) are provided. K KIND, W WH2, M GTBM globular tail domain binding motif, S SB SPIRE box, F FYVE-type zinc finger, H C-terminal flanking sequences similar to H2 of Slp/Slac-proteins, WB western blotting. Source data **b**–**f** are provided in the Supplementary Source data file.

---

(hereafter KW-Rab27a; Fig. 4a). Recent FRAP studies found that association of Rab27a with melanosomes is relatively stable indicating that the chimeric KW-Rab27a is likely to stably associate with melanosomes[50]. Accordingly, we found that KW-Rab27a, localised to melanosomes (unlike SPIRE2-KW) and restored their uniform dispersion in SPIRE1/2-depleted cells to a greater extent than SPIRE2-KW alone (Fig. 4a, b ((high magnification inset), Fig. 4c; mean % cells with uniformly dispersed melanosomes KW-Rab27a = 98.01 ± 2.0%). These findings indicate that association with Rab27a at the melanosome membrane enhances the ability of SPIRE proteins to uniformly disperse melanosomes, and supports the idea that differences in the Rab27a interaction affinity of SPIRE1 and SPIRE2 underlie different patterns of melanosome dispersal seen in cells expressing either protein alone.

**Membrane targeting of FMN formins reduces their function in melanosome transport.** Given the importance of FMN1/SPIRE interaction in melanosome transport and the finding that SPIRE proteins can associate with melanosomes, we next investigated whether FMN1 associates with melanosomes[29]. Confocal analysis of localisation in melan-f cells revealed that GFP-Fmn1 was distributed throughout the cytoplasm and not enriched adjacent to melanosomes, indicating that FMN1, unlike SPIRE1, does not strongly associate with melanosomes (Fig. 1f (inset)). This suggests that FMN1 may interact transiently, rather than stably, with melanosome-associated SPIRE proteins, as proposed by in vitro studies of AF assembly by SPIRE1 and Fmn2 (ref. [51]). To test this possibility, we used the Rab27a$^{SF1F4}$ mutant fusion strategy to generate the FH1-FH2-Rab27a fusion protein, which stably targets the FH1 and FH2 domains of FMN1 to melanosomes, and measured its activity in dispersing melanosomes in melan-f cells (Fig. 5a). We saw that although FH1-FH2-Rab27a localised to melanosomes and increased melanosome dispersion compared with GFP, it was significantly less effective than intact FMN1 or the FH1-FH2-FSI fragment (Fig. 5a, b (high magnification inset), Fig. 5c; mean pigment area (% total), FH1-FH2-Rab27a = 55.42 ± 17.49). This indicates that stable association with melanosomes reduces FMN1 function and supports idea that FMN1 transiently associates with melanosome-associated SPIRE1/2.

To further investigate the importance of the subcellular positioning of SPIRE1/2 and FMN1, we tested the ability of FMN1 related protein, FMN2, to rescue melanosome transport in melan-f cells. FMN2, like FMN1, contains FH1 and FH2 domains, and the C-terminus FSI, but unlike FMN1, FMN2 may be co-translationally myristoylated and targeted to the cell membrane[52,53]. Thus, FMN2 may generate AF near the cell

membrane, i.e., distant from the perinuclear clustered melanosomes in melan-f cells, rather than throughout the cytoplasm like FMN1 (Figs. 1f and 5b). We found that both intact FMN2 (hereafter GFP-FMN2) and the FMN2 FH1-FH2-FSI fragment expressed as C-terminus fusions to GFP were distributed throughout the cytoplasm, and restored melanosome dispersion with similar efficiency to GFP-FMN1 (Supplementary Fig. 10a–c; mean pigment area (% total); GFP-FMN1 = 84.11 ± 9.544, GFP-FMN2 = 81.72 ± 8.011 and GFP-FMN2 FH1-FH2-FSI = 86.42 ± 10.84). Meanwhile, although the N-terminus fusion of FMN2 to GFP (FMN2-GFP) dispersed melanosomes more efficiently than GFP alone, it did so with lower efficiency than GFP-FMN2 (Supplementary Fig. 10b, c; mean pigment area (% total); GFP = 44.17 ± 9.31, FMN2-GFP = 58.37 ± 7.818). To probe the role of N-myristoylation and membrane targeting in reducing the function FMN2-GFP versus GFP-FMN2, we blocked this by two approaches; (1) expression of the non-N-myristoylatable Fmn2-G2A mutant, and (2) treatment with the N-myristoyltransferase inhibitor (NMTi) IMP-1088 (refs. [53,54]). We found that both strategies redistributed FMN2-GFP to a cytoplasmic pattern, similar to GFP-FMN2 and GFP-FMN1, and enhanced its function in transport (Supplementary Fig. 10b, c; mean pigment area (% total); FMN2-[G2A]-GFP = 79.71 ± 11.72%, FMN2-GFP + NMTi = 79.03 ± 9.488%). Using a myristic acid analogue/ alkyne (YnMyr) labelling approach, we confirmed the N-myristoylation status of FMN2-GFP, and the effects of the G2A mutant and NMTi in blocking this modification (Supplementary Fig. 10d). These observations support the idea that Fmn1 does not stably associate with membranes in melanocytes.

## Discussion

Here we investigated the mechanism of myosin-Va-dependent melanosome dispersion in melanocytes, focusing on the role of dynamic AFs that we previously identified as essential for this process[15].

Our data indicate: (1) that FMN1 and SPIRE proteins regulate melanosome dispersal in melanocytes, (2) that FMN1 and SPIRE1/2 extend AFs from the membranes of melanosomes to generate a cell-wide network that links adjacent melanosomes and (3) that Rab27a can recruit SPIRE1/2 to melanosomes as effectors, along with myosin-Va and Mlph. The latter point raises the interesting possibility that Rab27a drives melanosome dispersion in melanocytes by co-ordinating the activity of AF motors (i.e., myosin-Va) and AF assembly machinery (i.e., SPIRE1/2 and FMN1).

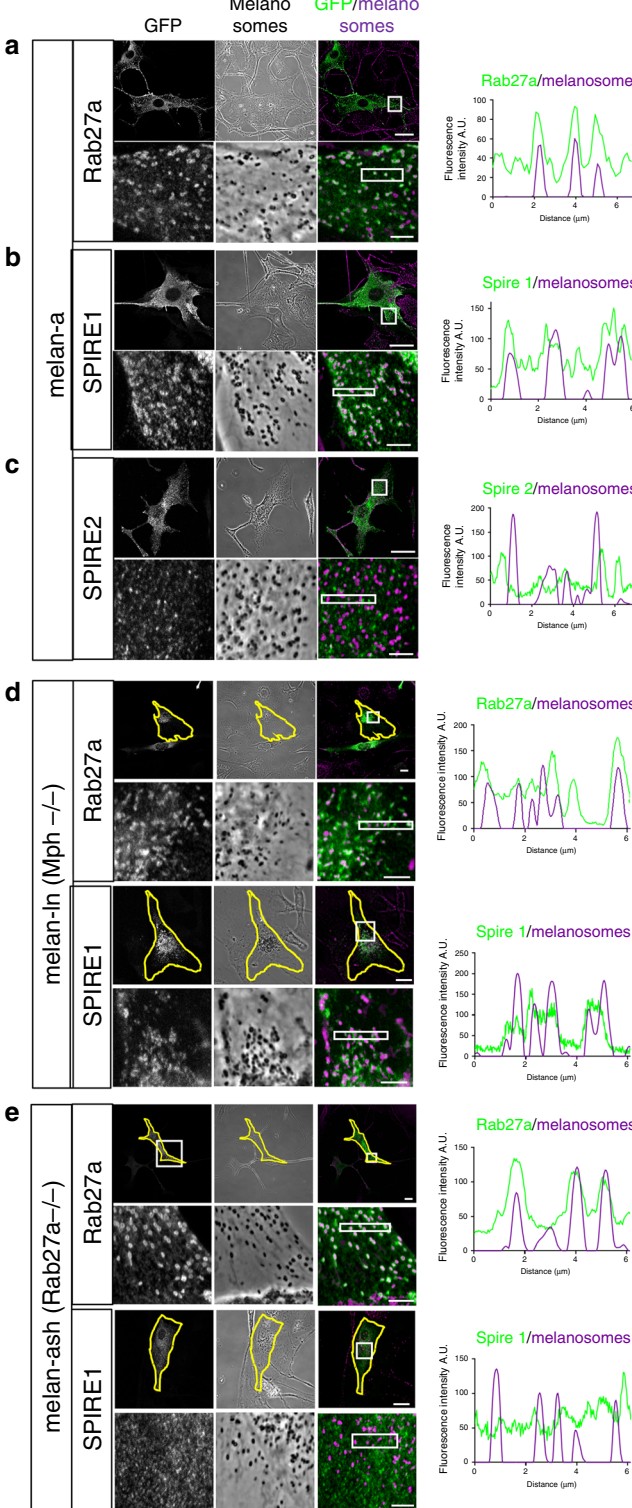

**Fig. 7 Rab27a recruits SPIRE1 to melanosomes in melanocytes.**
Melanocytes were transfected with plasmids allowing expression of the indicated proteins as fusions to the C-terminus of EGFP. Cells were fixed after 48 h, stained with GFP-specific antibodies to detect the expressed proteins, and the intracellular distribution of expressed protein and melanosomes was observed using a confocal microscope (see Experimental procedures). **a-e** Single confocal z-sections of the distribution of each protein, pigmented melanosomes (transmitted light/phase contract images) and merge images (melanosomes pseudo-coloured magenta). Upper panels show whole cells. Boxes indicate regions shown in lower panels at high magnification allowing comparison of the distribution of melanosomes and fluorescent protein. Line plots are fluorescence intensity profile plots of the boxed regions in high-magnification images averaged along the vertical axis. **a-c**, **d** and **e** are melan-a, melan-ln and melan-ash cells. **d**, **e** Yellow lines indicate the borders of transfected cells. Scale bars = 20 μm and 3 μm in main images and magnified portions. Source data are provided in the Supplementary Source data file.

Mlph and Slp2 (refs. [41,49]). Interaction with these effectors obscures large portions of the surface of Rab27, including the switch regions, meaning that it is unlikely that a single Rab27 protein could interact with both effectors simultaneously. Rab27a interaction may activate SPIRE1/2 at the melanosome membrane by disrupting the intramolecular regulatory KIND/FYVE interaction[35]. Active SPIRE1/2 may then transiently recruit FMN1, via SPIRE1/2-KIND:FMN1-FSI interaction, thereby allowing FMN1 with SPIREs to associate with, and elongate, the +/barbed ends of short AFs located at the melanosome membrane. This would result in the formation of an AF network that is polarised with +/barbed ends oriented away from the melanosome membrane (i.e., towards other melanosomes and the cell periphery; Fig. 9b). This is consistent with the observation that melan-a, but not melan-f or SPIRE1/2 depleted, melanocytes harbour a population of AFs that emanate from, and link adjacent melanosomes. Thus myosin-Va attached to an adjacent melanosome may then 'walk' towards the +/barbed ends of one of these AFs thereby dispersing melanosomes from one another (Fig. 9c).

Extrapolating from this two melanosome scenario to the cellular level, we propose that this pattern of myosin-Va/AF-dependent inter-organelle repulsion could drive and sustain dispersal of melanosomes from one another, resulting in their dispersal throughout the cytoplasm, as seen in wild-type melanocytes (Fig. 9d). We suggest that such an inter-connected, three-dimensional network of organelles, tracks and motors could allow cell-wide coupling of local organelle-associated force generators (i.e., myosin-Va motors and FMN1/SPIRE1/2 AF assembly machinery). And that this arrangement could allow the generation of sufficient force to rapidly disperse these large, rigid and numerous organelles (1000 s/cell), and maintain this distribution, more efficiently than conventional transport models, which envisage single organelles pulled along a limited number of single tracks by motors independently of other cargo, motors and tracks.

As a preliminary to test of our model, we used the open-source modelling framework Cytosim to simulate melanosome dispersion by myosin-Va motors and FMN1/SPIRE1/2 AF builders. Based on our data, we modelled AFs polarised with their −/pointed ends attached (via Rab27a/SPIRE interaction) to melanosomes and +/barbed ends growing (extended by FMN1) into the cytoplasm, and myosin-Va anchored with their cargo-binding tail (and Rab27a/Mlph) to the melanosome and motor domains free in the cytoplasm (hereafter termed 'melanosim-attached', see Supplementary Code 1). Running the simulation

Based on these findings, we propose a model by which this might occur (Fig. 9). Firstly, we suggest that Rab27a-GTP recruits two effector complexes to melanosomes; (i) myosin-Va/melanophilin and (ii) SPIRE1/2 (Fig. 9a). The recently identified myosin-Va:SPIRE1/2 interaction may also stabilise SPIRE1/2 at the melanosome membrane and/or integrate SPIRE1/2 activity with myosin-Va[38]. We expect that the recruitment of two effector complexes requires two pools of Rab27a, as the Rab27 interaction domain of SPIRE1/2 appears conserved with other effectors, e.g.,

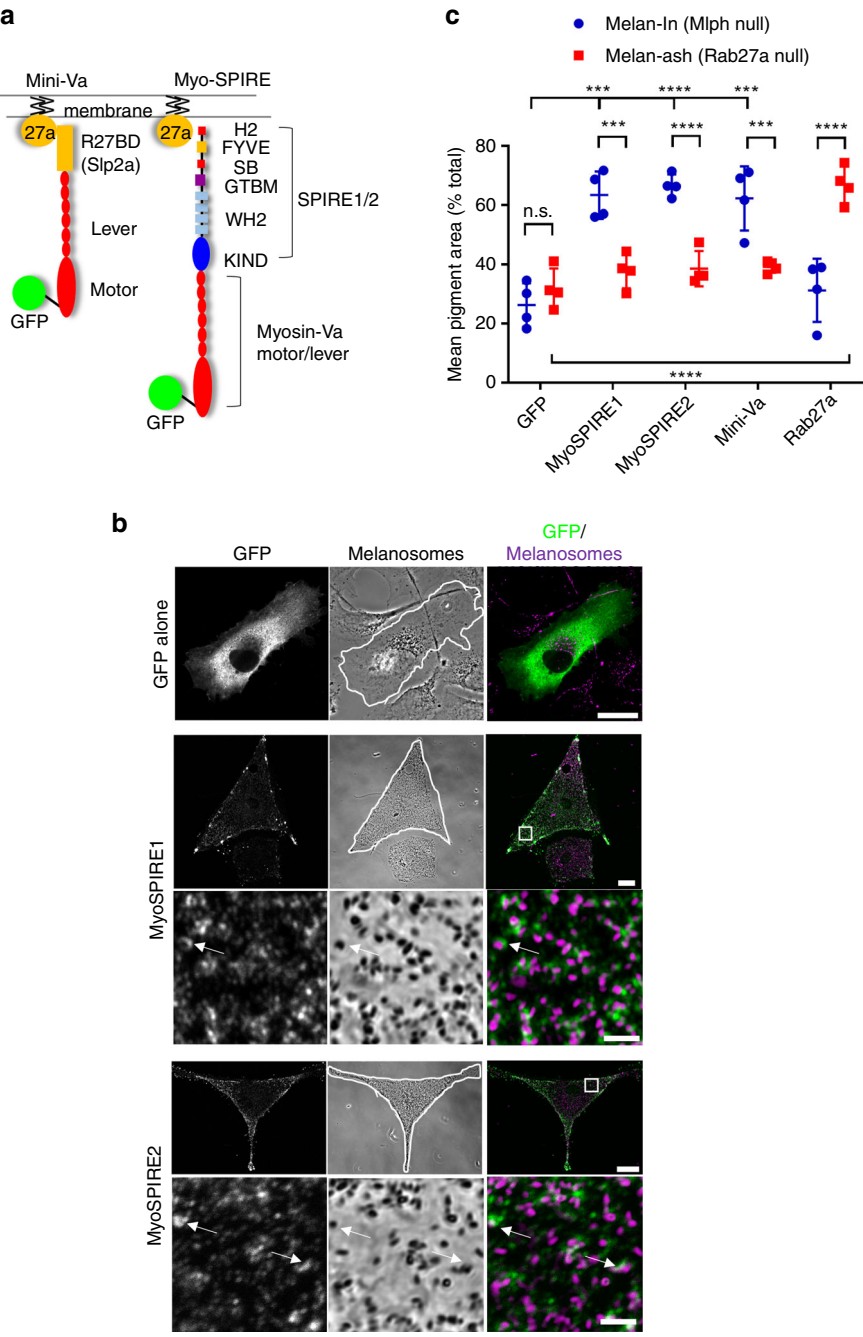

**Fig. 8 Functional evidence that SPIRE1/2 interact with melanosome-associated Rab27a in melanocytes.** melan-In (Mlph null) and melan-ash (Rab27a null) melanocytes were infected with viruses expressing the indicated proteins. Cells were fixed after 24 h, stained with GFP-specific antibodies, and the intracellular distribution of GFP-fusion proteins and melanosomes was observed using a confocal microscope (see Experimental procedures). **a** A schematic representation of the structure of mini-Va and myoSPIRE1/2 proteins, and their interaction with membrane-associated endogenous Rab27a. R27BD Rab27-binding domain, SB SPIRE box, GTBM globular tail binding domain. **b** Single confocal z-sections showing the distribution of expressed proteins, melanosomes and their colocalisation (from left to right). In merge images melanosomes are false-coloured magenta. For myoSPIRE1/2 upper and lower panels are low and high-magnification images. Boxes in low-magnification images indicate the area presented in the high-magnification images below. Scale bars = 10 μm in main images and 2 μm in magnified portions. Arrows indicate colocalisation of spots of myoSPIRE with melanosomes. Cell outlines are shown by white lines in phase contrast (melanosome) images. **c** A bee swarm plot showing the effect of expression of myoSPIRE and other proteins on melanosome distribution in melanocytes (melan-In/Mlph null and melan-ash/Rab27a null). Data presented are mean pigment area measurements from four independent experiments. In each case, pigment area was measured for ten cells expressing each of the different proteins. ****, *** and n.s. indicate significant difference $p = <0.0001$, 0.001 and none between the datasets linked by horizontal bars as determined by one-way ANOVA. Bars within datasets indicate the mean and 25th and 75th percentile of data. Source data for **c** are provided in the supplementary Source data file.

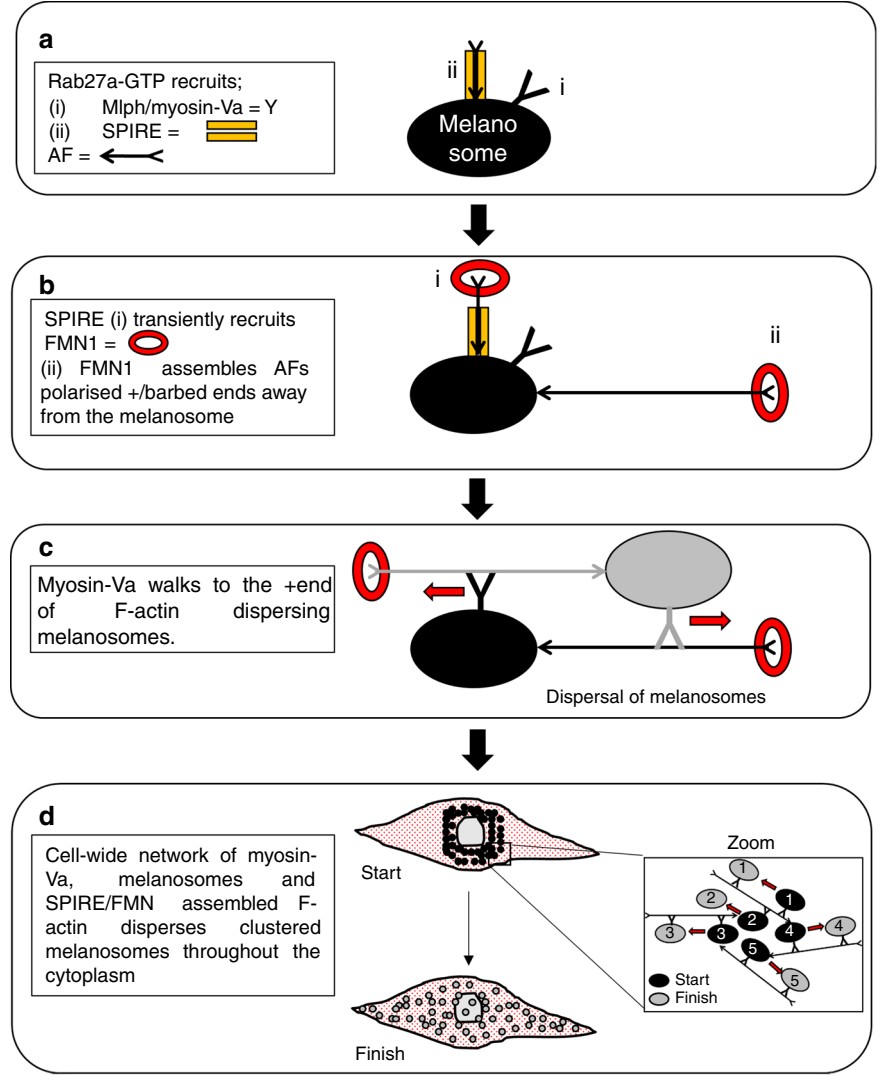

**Fig. 9 A model indicating how Rab27a could regulate AF-driven melanosome transport by integrating the activity of myosin-Va, and SPIRE1/2 and FMN1.** See main text for details. In **a**, a model melanosome, myosin-Va, SPIRE and an actin filament are shown, Rab27a is not shown for clarity but is present on the melanosome membrane. In **b**, the activity of FMN1 in extending the +end of melanosomes from melanosomes is shown. In **c**, grey and black colours indicate AF and myosin-Va associated with melanosomes of the same colour. In **d**, grey and black colours indicate the position of numbered melanosomes at the start and finish of an episode of dispersive transport.

from a starting point, in which melanosomes were clustered in the centre of the model cell, we saw that as AFs emerged from melanosomes myosin-Va motors on adjacent melanosomes transiently attached to the filaments and the linked melanosomes moved apart relative to one another (Supplementary Fig. 11a, Supplementary Movies 1 and 2, see Supplementary Fig. 11b for examples). This resulted in rapid global dispersion of the clustered population of melanosomes throughout the cytoplasm of the cell, as predicted by our mode. This closely resembled previous live-cell observation of melanosome dispersion in melan-d cells, in response to acute activation of a pharmacologically activatable myosin-Va motor[15]. The extent of dispersion was reduced in simulations using free AFs whose pointed ends were not attached to melanosomes ('melanosim-detached', Supplementary Fig. 11a, Supplementary Movie 3). These observations support the hypothesis that a melanosome-associated Rab27a can generate a network of organelle linked motors and filament builders, and disperse rapidly

and uniformly organelles throughout the cytoplasm of a eukaryotic cell.

Several pieces of data presented here, and previously, support this possibility. Firstly, melan-f cells and SPIRE1/2 depleted melan-a cells show significantly fewer melanosome-associated AFs compared with melan-a cells. Secondly, fusion of active SPIRE2 and FMN1 to melanosomes via the Rab27a$^{SF1F4}$ mutant boosts the function of SPIRE2-KW and reduces the function of FMN1-FH1-FH2, consistent with stable and transient modes of association of SPIRE1/2 and FMN1 with melanosomes. Thirdly, consistent with the predicted requirement for proximity between melanosomes and AFs, re-targeting of FMN (FMN2-GFP) to the cell membrane via N-myristoylation reduces its function. Fourthly, we previously found that the processive +/barbed end directed transport activity of myosin-Va was essential for its function in melanosome dispersion[15].

Beyond melanocytes evidence of such non-conventional or network-like organisations of the acto-myosin cytoskeleton

exist in other cell types, e.g., the sarcomere of muscle cells and stress fibres of migrating fibroblasts represent. These structures allow the amplification of force generated by myosin-II motors to drive cell shape changes and movement[55,56]. Meanwhile recent work in oocytes reveal that myosin-Vb, SPIRE1/2 and FMN2 form a meshwork of short AFs that drive rapid, long-range outward flow of a large population of Rab11-positive vesicles to the plasma membrane, independently of MTs (refs. [23,57]). With our data, this suggests that similar AF-dependent organelle transport systems might be widespread in somatic mammalian cells.

Our work additionally highlights differences between SPIRE1 and SPIRE2 function. In particular, we found that SPIRE1 has higher affinity for interaction with Rab27a, localised more strongly to melanosomes and more efficiently drove uniform melanosome dispersion compared with SPIRE2. In contrast uniform melanosome dispersion required higher levels of SPIRE2 expression and lower levels resulted in hyper-dispersion. The KW-Rab27a fusion protein uniformly dispersed melanosomes more efficiently than SPIRE2-KW, conversely SPIRE1/2 mutants compromised in membrane/Rab27a interaction promoted hyper-dispersion. This suggests that interaction with Rab27a is important in ensuring uniform dispersion, as seen in wild-type mela-nocytes, by toning down and localising the hyper-dispersive activity of the KW to melanosomes. In parallel, we found that the SPIRE1 gene was more highly expressed than SPIRE2 and that single depletion of SPIRE1, but not SPIRE2, reduced dispersion leading to perinuclear clustering and hyper-dispersion of mela-nosomes. Overall, these data suggest that SPIRE1 plays a more significant role in melanosome transport than SPIRE2. Currently the mechanism by which SPIRE2 hyper-disperses melanosomes at low expression levels is unclear. One possibility is that at low concentration SPIRE2 is activated by interaction with factors e.g. other Rabs, associated with peripherally located compartments, and assembles dynamic AFs at the periphery that allow local accumulation of melanosomes. Meanwhile at higher concentrations SPIRE2 may be activated by Rab27a on melano-somes triggering uniform melanosome dispersal via the mechanism proposed in Fig. 9. Future studies should address this question.

Finally, we suggest that our data have broader implication for transport in the many other cell types in which Rab27, related Rab proteins, SPIRE1/2, FMN and myosin-V proteins are expressed. In particular we note the similarity between the proteins identified here and those shown to regulate the vesicle associated AF meshwork that drives asymmetric spindle posi-tioning during meiosis in mouse oocytes (SPIRE1/2, FMN2, Rab[11] and myosin-Vb)[22,23]. The finding that similar groups of proteins drive similar activities i.e. rapid, long-range AF-dependent organelle transport, in different cell types suggests that related machineries may drive transport in other animal cell types. Consistent with this in mammals Rab27 regulates the dispersal and secretion of organelle contents in many specia-lised 'secretory' cell types e.g. inflammatory mediator contain-ing granules in mast cells, lytic granules in cytotoxic lymphocytes and exosome containing MVBs in cancer cells[58–60]. The oligodactylism phenotype of the FMN1 knockout mouse also supports the possible function of FMN1 in secretory transport processes, since proper limb development requires concerted communication between cells, which is vitally dependent on secreted proteins[25]. Also, the behavioural phe-notypes of FMN2 and SPIRE1 mutant mice may be caused by altered secretion of neuropeptides or growth factors[61,62]. Thus, it will be interesting to test whether Rab27a contributes simi-larly to co-ordinate the AF-dependent transport of other organelles in other cell types.

## Methods

**Animal procedures**. All animal procedures with mice reported in this work were first validated by the National Centre for Biotechnology Ethics Committee on Animal Experimentation, thereafter approved by the National CSIC Ethics Com-mittee and eventually authorised by the Autonomous Community of Madrid, acting as the competent authority, according to the Spanish legislation (L32/2007, RD53/2013, L6/2013, OM ECC/566/2015 and RD1386/2018) and the European Directive 2010/63/EU, with the approval reference PROEX 343/15. All mice were housed at the registered CNB animal facility, fed and provided water and regular rodent chow ad libitum, with a light/dark cycle 08:00–20:00, according to the European and Spanish norms, and the animal welfare recommendations.

**Derivation and maintenance of cultured cells**. Cultures of immortal FMN1-deficient melanocytes (melan-f) were derived essentially[63]. In brief, mice carrying a previously generated FMN1 loss of function allele (FMN1pro) were crossed with Ink4a-Arf mutant mice, in order to generate pups homozygous for the FMN1pro mutant allele and heterozygous for Ink4a-Arf mutant allele. Genotyping of the embryos was carried out by PCR screening[25,63]. Melanocytes were then derived from the dorsal skin of newborn mutant mice[64]. Cultures of immortal melan-f, melan-ash, melan-ln and melan-a melanocytes, and HEK293a were maintained, infected with adenovirus expression vectors and transfected with siRNA oligonu-cleotides at a final concentration of 66.7 fmol/µl of transfection mixture and Oli-gofectamine transfection reagent diluted 300-fold in optiMEM medium (both Thermo-Fisher)[65,66]. The melanocyte cell lines described here are available from the Wellcome Trust Functional Genomics Cell Bank http://www.sgul.ac.uk/depts/anatomy/pages/WTFGCB.htm.

**Transfection of cultured cells**. Cultured cells were transfected with siRNA as described above and plasmid DNA at a final concentration of 1.67 ng/µl using Fugene 6 transfection reagent (producer) diluted 300-fold[66]. The sequence of siRNA oligonucleotides used in this study are are indicated in Supplementary Table 4. siRNA oligonucleotides were purchased from Dharmacon Thermo-Fisher and Sigma Genosys UK.

**Quantitative real-time PCR**. Primers and the probes for Q-RT-PCR targets (from Sigma Genosys, Cambridge, UK) were designed using Primer Express software (Life Technologies). Probes were labelled at the 5′- and 3′-ends with fluorophore 6-FAM (6-carboxyfluorescein) and quencher TAMRA (tetramethylrhodamine), respectively (Supplementary Table 5). To generate mRNA samples, pools of melan-a cells grown in six-well plates ($1 \times 10^5$/well) were transfected with siRNA in triplicate as described previously[66]. Seventy two hours later cells were harvested and mRNA extracted using the RNeasy Mini RNA extraction kit (Qiagen). cDNA was generated using Moloney Murine Leukaemia Virus M-MLV reverse tran-scriptase (Promega) using random primers. To generate a standard curve of signal: template concentration for each Q-RT-PCR assay a pool containing 5% of each of the cDNA samples analysed was generated. This pool was serially diluted in DEPC water (1:4, 1:16, 1:64, 1:256) and these were used as template in target gene and GAPDH Q-RT-PCR assays. (The shape of the standard curve indicates the rela-tionship between signal and template concentration. For all assays standard curves gave straight lines with $R^2 > 0.99$, indicating that there is a linear relationship between signal and template). To measure the expression of targets in siRNA-transfected cells each neat cDNA was diluted 1:32 in DEPC water and the following reagents were added per well of a 96-well plate: 6.5 µl TaqMan Fast 2× PCR Master Mix (Life Technologies); 0.4 µl forward primer (10 µM); 0.4 µl reverse primer (10 µM); 0.25 µl probe (10 µM); 3 µl cDNA; 2.45 µl DEPC water. For each sample, three technical repeats were performed. Reaction plates were sealed with optically clear adhesive film, centrifuged, and Q-RT-PCR performed using a StepOnePlus Real-Time PCR system (Applied Biosystems) using the 'fast' mode. CT values for each reaction were determined by the StepOne software. The slope (S), intercept (I) and $R^2$ values were calculated for the standard curve of each Q-RT-PCR assay. CT values from siRNA-transfected samples were then processed to generate a 'quantity value' for each CT value as follows: (1) $(CT-I)/S = LQ$, (2) $10^{LQ} = Q$, (3) $Q \times (1/MNT) = GOIP$ (where MNT = mean non-targeted quantity value) and (4) $GOIP/GP$ = normalised expression of target relative to GAPDH (where GP is the nor-malised quantity value for the GAPDH primer). Expression of FMN1 was detected using Taqman assay Mm01327668_m1 from Thermo-Fisher Scientific, UK.

For measurement of the expression levels of SPIRE, FMN and MLPH genes, the RNA isolation kit of Macherey-Nagel (Düren, Germany) was used. To reduce amount of melanin and increase purity of total RNA step two of the protocol was repeated. In order to quantify the amount of total RNA isolated, spectrophotometric determination was used. cDNA was generated employing the Qiagen QuantiNova® Reverse Transcription Kit (according the manufacturer's protocol). An internal control RNA was used to verify successful reverse transcription. PCR primer sets were designed using primer3, Primer-Blast and were constructed by Sigma-Aldrich (Taufkirchen, Germany). The primer sets and templates used are shown in Supplementary Table 6. For primer testing generated cDNA from melan-a total RNA and plasmid DNAs as positive controls were used as templates. For absolute quantification, an external plasmid-cDNA standard was used to create a standard curve with known copy number concentrations.

Linearised plasmid DNA was serially diluted in water. The copy number of standard DNA molecules could be determined by the following formula: $((X \text{ g/µl DNA})/[\text{plasmid length in basepairs} \times 660]) \times 6.022 \times 10^{23} = Y$ molecules/µl). The absolute quantification by Q-RT-PCR was performed with the Rotor-Gene Q (Qiagen) thermocycler. For each reaction triplets were performed, and CT values were determined with the help of the Qiagen Rotor-Gene Q Series Software. CT values of the samples are compared to the standard curve with known concentrations to calculate the amount of target in the samples. For this purpose, the QuantiNova® SYBR® Green PCR Kit (GE Healthcare Life Sciences, Freiburg, Germany) and the associated protocol were employed.

**Immunoblotting**. SDS–PAGE was carried out using 4–15% gradient SDS–PAGE gels (Bio-rad #4561085) and a Mini-PROTEAN Tetra Cell (Bio-rad 1658000EDU). Molecular weight standards were BLUeye Prestained Protein Ladder (Sigma-Aldrich 94964-500UL). Proteins were and transferred to PVDF membrane (Merck Millipore HVLP04700) using a Hoefer™ TE22 Mini Tank Blotting Unit (Fisher Scientific 10380895)[38,66]. Details of primary and secondary antibodies used in this study are in Supplementary Tables 7 and 8. Uncropped western blots are available in the Supplementary Information file.

Signal was detected using a Li-Cor infrared scanner (Odyssey; Fig. 1d, Supplementary Fig. 4) or by chemiluminescence (Luminata Forte Western HRP substrate; Merck Millipore, Darmstadt, Germany; Fig. 6, Supplementary Figs. 5 and 6) recorded with an Image Quant LAS4000 system (GE Healthcare Life Sciences; Supplementary Fig. 10b).

**Multiple protein sequence alignments and phylogenetic tree generation**. Multiple protein sequence alignments for C-terminal regions of human SPIRE1 and SPIRE2, and N-terminal regions of human MLPH, MYRIP, Rabphilin-3A and NOC2 amino acid sequences, were performed using the Clustal Omega Multiple Sequence Alignment tool (EMBL-EBI, Hinxton, UK). Respective cDNA sequences were obtained from the NCBI Gene database with the following accession numbers: Homo sapiens (Hs)-SPIRE1: NM_020148.2, Hs-SPIRE2: AJ422077, Hs-MLPH: NM_024101.6, Hs-MYRIP: NM_001284423.1, Hs-RPH3A (Rabphilin-3A): NM_001143854.1, Hs-RPH3AL (NOC2): NM_006987.3. Generation of phylogenetic trees for Rab27 interacting proteins was based on multiple amino acid sequence alignments using Clustal Omega as described above and manual confinements. Respective cDNA sequences were obtained from the NCBI Gene database with the following accession numbers: Hs-RIMS1: NM_014989.5, Hs-RIMS2: NM_001100117.2, Hs-RPH3A (Rabphilin-3A): NM_001143854.1, Hs-RPH3AL (NOC2): NM_006987.3, Hs-MLPH (SLAC2-A): NM_024101.6, Hs-EXPH5 (SLAC2-B): NM_015065.2, Hs-MYRIP (Slac2-c): NM_001284423.1, Hs-SYTL1 (SLP1): XM_005246022.3, Hs-SYTL2 (SLP2-A): NM_032943.4, Hs-SYTL3 (SLP3-A): NM_001242384.1, Hs-SYTL4 (SLP4, granuphilin): NM_080737.2, Hs-SYTL5 (SLP5): NM_001163334.1, Hs-SPIRE1: NM_020148.2, Hs-SPIRE2: AJ422077. Evolutionary analyses and phylogenetic tree formation were conducted in MEGA7 (ref. [67]). The evolutionary history was inferred by using the Maximum Likelihood method based on the JTT matrix-based model[68]. The bootstrap consensus tree inferred from 500 replicates is taken to represent the evolutionary history of the taxa analysed[69]. Branches corresponding to partitions reproduced in <50% bootstrap replicates are collapsed. The percentage of replicate trees in which the associated taxa clustered together in the bootstrap test (500 replicates) are shown next to the branches[69]. Initial tree(s) for the heuristic search were obtained automatically by applying Neighbour-Join and BioNJ algorithms to a matrix of pairwise distances estimated using a JTT model, and then selecting the topology with superior log likelihood value. The analysis involved[14] amino acid sequences. There was a total of 188 positions in the final dataset.

**Generation of bacterial and mammalian protein expression vectors**. Expression vectors were generated by standard cloning techniques using AccuPrime Pfx (Thermo-Fisher, Waltham, MA, USA) or Pfu (Promega, Mannheim, Germany) DNA polymerases, restriction endonucleases and T4 DNA ligase (both New England Biolabs, Frankfurt am Main, Germany). Point mutants were generated using the In-Fusion HD cloning kit (TakaraBio/Clontech). DNA sequencing was carried out by LGC Genomics (Berlin, Germany) and Source Bioscience (Nottingham, UK). Supplementary Table 9 shows details of vectors used in this study. Amino acid boundaries are related to the following cDNA sequences: human SPIRE1 (NM_020148.2), human SPIRE2 (AJ422077), murine FMN1 (XP_011237597), murine FMN2 (NP_062318.2), murine myosin-Va (XM_006510827.3) and murine Mlph (NP_443748.2). To generate mRuby3 fusion protein encoding expression vectors, the pKanCMV-mClover3-mRuby3 vector was used as template to PCR amplify the mRuby3 cDNA sequence for further subcloning. This vector was a gift from Michael Lin and was provided by Addgene (plasmid #74252)[70].

**Recombinant protein expression and purification**. Recombinant GST-Rab27a-WT, Q78L mutant and T23N mutant proteins were expressed in *Escherichia coli* Rosetta bacterial cells (Merck Millipore, Novagen). Bacteria were cultured in LB medium (100 mg/l ampicillin, 34 mg/l chloramphenicol) at 37 °C until an A600 nm of OD 0.6–0.8. Protein expression was induced by 0.2 mM isopropyl-β-D-

thiogalactopyranoside (Sigma-Aldrich, Taufkirchen, Germany) and continued at 16–20 °C for 18–20 h. Bacteria were harvested and lysed by ultra-sonication. Soluble proteins were purified by an ÄKTApurifier system (GE Healthcare Life Sciences) using GSH-Sepharose 4B (GE Healthcare Life Sciences) beads and size-exclusion chromatography (High Load 16/60 Superdex 200; GE Healthcare Life Sciences). Proteins were concentrated by ultrafiltration using Amicon Ultra centrifugal filters (Merck Millipore) with respective molecular weight cut offs. The final protein purity was estimated by SDS–PAGE and Coomassie staining.

**Nucleotide loading of GST-Rab27a**. In order to analyse the interaction of GST-Rab27a and SPIRE1 depending on its nucleotide loading (GTP versus GDP), GST-Rab27a proteins were loaded with the non-hydrolysable GDP (GDPβS) and GTP (GTPγS) analogues (both from Sigma-Aldrich), respectively, prior to GST pull-down assays. For each nucleotide exchange 5.8 mg of purified GST-Rab27a fusion protein was used, corresponding to 2.5 mg pure Rab27a protein, and mixed with a fourfold molar excess of GDPβS and GTPγS (1.13 mM each), respectively, 12.5 units alkaline phosphatase (CIAP; Roche, Penzberg, Germany) and CIAP buffer (20 mM Tris-HCl pH 7.4, 200 mM NaCl, 1 mM MgCl₂, 2 mM DTE, 200 mM (NH₄)₂SO₄, 100 µM ZnCl₂) in a total volume of 800 µl. The mixture was incubated for 17 h at 4 °C on a rotating wheel. On the next day, CIAP buffer was exchanged by RAB polarisation buffer (20 mM Tris-HCl pH 7.4, 200 mM NaCl, 5 mM MgCl₂, 1 mM DTE, 5 µM GDPβS/GTPγS) using NAP-10 columns (GE Healthcare Life-sciences) according to manual.

**GST pull-down from HEK293 cell lysates**. HEK293 cells were cultured and transfected with plasmid DNA as described previously[38]. To ensure equal input of GFP-tagged or Myc-epitope-tagged SPIRE1 and SPIRE2 deletion mutants for GST pull-downs, all SPIRE mutants were first expressed in HEK293 cells and respective protein expression levels were analysed by western blotting (anti-GFP, anti-c-Myc). Protein bands were quantified using ImageQuant TL (GE Healthcare Life Sciences) and normalised to the lowest signal band. According to quantification, the amount of cell lysates employed in subsequent pull-downs was adjusted. For pull-downs, HEK293 cells were transfected with expression vectors encoding fluorescent protein or Myc-epitope-tagged proteins. Forty eight hour post transfection, cells were lysed in lysis buffer (25 mM Tris-HCl pH 7.4, 150 mM NaCl, 5 mM MgCl₂, 10% (v/v) glycerol, 0.1% (v/v) Nonidet P-40, 1 mM PMSF, protease inhibitor cocktail) and centrifuged at $20{,}000 \times g$, 4 °C, 20 min to remove insoluble debris. For GST pull-down assays 50 µg GST-RAB27A (-WT, -Q78L, T23N, -GTPγS, -GDPβS) proteins and 25 µg GST control was coupled to GSH-Sepharose 4B beads (1:1 suspension) for 1 h, 4 °C on a rotating wheel. Beads were washed twice with pull-down buffer (25 mM Tris-HCl pH 7.4, 150 mM NaCl, 5 mM MgCl₂, 10% (v/v) glycerol, 0.1% (v/ v) Nonidet P-40) and subsequently incubated with cell lysates for 2 h at 4 °C on a rotating wheel. Here, 1 ml lysates with the lowest expressed protein was employed and lysates with higher protein abundance were diluted with pull-down buffer to 1 ml, according to prior quantification. Beads were washed four times with pull-down buffer and bound proteins were eluted with 1× Laemmli buffer, denatured at 95 °C for 10 min, and subsequently analysed by immunoblotting.

**Quantitative GST pull-down assays**. GST pull-down assays were performed as described above from HEK293 cell lysates with increasing concentrations of GST-Rab27a-Q78L protein. According to prior quantification, the concentration of ectopic expressed GFP-SPIRE1-MSFH was ~50 nM. Cell lysates were pooled and equally distributed to beads coupled GST-Rab27a-Q78L protein. Beads were pelleted and the cell lysate supernatant was diluted 1:4 with aqua. dest. Each sample was allowed to adapt to 20 °C for 10 min in a water bath. The concentration-dependent binding of Rab27a-Q78L to C-terminal GFP-SPIRE1-MSFH was determined by fluorospectrometric analysis using FluoroMax- 4 Spectrofluorometer (Horiba Jobin Yvon, Bensheim, Germany). The same experiment was performed employing GFP-Mlph-RBD expressed in HEK293 cells and purified GST-Rab27a-Q78L proteins. The AcGFP1 green fluorescent protein was excited at 488 nm (slit = 5 nm) and the emission at 507 nm was recorded (emission maximum, slit = 5 nm) with an integration time of 0.1 s. The data were calculated as fraction bound (%) ($y$) using the equation (1) $y = ((y_o - y_c)/y_o) \times 100$, where $y_o$ and $y_c$ equal the fluorescence signal of the supernatant of pull-down experiments carried out in the absence and presence of GST-Rab27a-Q78L protein coupled to beads.

Furthermore, data were analysed in SigmaPlot 12.3 software (Systat Software, Erkrath, Germany). Equilibrium binding data were fitted according to the equation (2) $y = (B_{max} \times x)/(K_d \times x)$ with $B_{max}$ representing the maximal amplitude, $K_d$ representing the equilibrium constant and $x$ representing the concentration of GST-Rab27a-Q78L.

**Microscopy and image analysis**. Cells ($1 \times 10^4$) for immunofluorescence were cultured on 13 mm diameter 1.5 thickness glass coverslips (SLS, Nottingham, UK. 4616-324139) for at least 24 h, prior to infection with adenovirus or transfection with plasmid DNA. Twenty-four hours later cells were paraformaldehyde fixed, stained, and fluorescence and transmitted light images were then collected using a Zeiss LSM710 confocal microscope fitted with a 63× 1.4NA oil immersion Apochromat lens or a Zeiss Axiovert 100 S inverted microscope fitted with 10× and 40×

objective lenses, and an Axiocam MR3 CCD camera. Primary and secondary antibodies used in this study are listed in Supplementary Tables 7 and 8, F-actin was detected using texas-red-phalloidin (Sigma P1951; 100 nM).

**Analysis of melanosome dispersion/transport**. For analysis of the effects of siRNA on melanosome distribution transfections were carried out in triplicate, i.e., three wells of a 24-well plate for each siRNA in each experiment. Seventy two hours later phase contrast images of three different randomly selected low power (using 10× objective) fields of cells in each well were captured using Axiovision 4.8 software associated with a Zeiss Axiovert 100 S inverted microscope. Images were then randomised and the number of cells with clustered melanosomes in each field was counted by a researcher blinded to the identity of the siRNA transfected into each field of cells. Cells in which pigmented melanosomes were contained within the perinuclear cytoplasm (<50% of the total cytoplasmic area) were defined as containing clustered melanosomes. Measurement of the function of adenovirus expressed FMN1 and SPIRE1/2 proteins in melanosome transport was based on manual measurement of the proportion of cell area occupied by pigmented melanosomes[65].

**Bimolecular fluorescence complementation**. For BiFC HEK293a cells were transfected with plasmids allowing expression of mCherry (control for transfection efficiency), vYNE-SPIRE1/2 (human SPIRE1/2 fused to the C-terminus of enhanced vYFP (accession CAO91538.1) N-terminus fragment (amino acids 1–173)) and vYCE-Rab27a wild type and mutants (Rab27a (rat)) fused to the C-terminus of enhanced vYFP C-terminus fragment (amino acids 155–239) using Fugene 6 transfection reagent (Promega E2691). Forty-eight hours later mCherry and YFP (BiFC) fluorescence was recorded from living cells in low power fields of view for each condition using the Zeiss Axiovert 100 S microscope, using the same exposure setting for each channel and each condition. Normalised BiFC signal was determined using ImageJ software. Briefly mCherry images were converted to binary and ROIs corresponding to transfected cells were saved to the regions manager. Mean vYFP for each ROI was extracted and normalised BiFC determined for each ROI by dividing the vYFP signal by the corresponding mCherry signal to normalise for differences in the level of transfection in each cell. Median BiFC signal for ROI was determined using Graphpad Prism[7] software (Graphpad, La Jolla, USA).

**Colocalisation analysis**. Fixed cells were analysed with a Leica AF6000LX fluorescence microscope, equipped with a Leica HCX PL APO 63×/1.3 GLYC objective and a Leica DFC350 FX digital camera (1392 × 1040 pixels, 6.45 μm × 6.45 μm pixel size; all from Leica, Wetzlar, Germany). 3D stacks were recorded and processed with the Leica deconvolution software module. Images were recorded using the Leica LASX software and further processed with Adobe Photoshop, and subsequently assembled with Adobe Illustrator. The extent of colocalisation of myosin-Va, Rab27a and SPIRE1 proteins at intracellular melanocyte membranes, as well as the localisation of Rab27a and SPIRE1 C-terminal fragments, respectively, at melanosome membranes was analysed using the ImageJ (V2.0.0) plug-in Coloc2. Here, the colocalisation rate is indicated by the Pearson's correlation coefficient (PCC) as a statistical measure to unravel a linear correlation between the intensity of different fluorescent signals. A PCC value of 1 indicates a perfect colocalisation, 0 indicates a random colocalisation and a PCC value of −1 indicates a mutually exclusive localisation of the analysed signals. To take the noise of each image into account and to gain an objective evaluation of PCC significance, a Costes significance test was performed. To do so, the pixels in one image were scrambled randomly and the correlation with the other (unscrambled) image was measured. Significance regarding correlation was observed when at least 95% of randomised images show a PCC less than that of the original image, meaning that the probability for the measured correlation of two colours is significantly greater than the correlation of random overlap[71,72].

**Cytoskeleton preparations for FESEM**. melan-a and melan-f cells were plated on 7 mm glass coverslips and grown for 24 h. The medium was aspirated and the cells were extracted for 1 min with 0.25% Triton X-100 (Sigma, UK) and 5 μM phallacidin (Sigma) in cytoskeletal stabilisation buffer (CSB; 5 mM KCl, 137 mM NaCl, 4 mM NaHCO₃, 0.4 mM KH₂PO₄, 1.1 mM Na₂HPO₄, 2 mM MgCl₂, 5 mM Pipes, 2 mM EGTA and 5.5 mM glucose, pH 6.1; ref.[73]). After a quick wash in CSB, the samples were fixed with 1% glutaraldehyde (Sigma) in CSB with 5 μM phallicidin for 15 min. The coverslips were transferred to HPLC-grade water and incubated with 2% osmium tetroxide in water for 1 hour. After three 15 min washes in distilled water, the preparations were dehydrated through successive immersion in 30, 50, 70, 90 and 100% ethanol solutions (three times 10 min each), followed by two 20 min incubation in methanol. The coverslips were carefully picked from the methanol solution and gently immersed vertically in HMDS (Hexamethyldisilazane, Sigma) in glass bottles for 30 s twice and left to air-dry for at least an hour before mounting them on 10 mm diameter specimen stubs. The coverslips were then coated with silver paint and sputter coated with a 5–6 nm layer of platinum an Edwards S150B sputter coater. Samples were imaged in a JEOL JSM-6700F scanning electron microscope by secondary electron detection. For phalloidin labelling of AFs, melan-a cells were permeabilised for 45 s with 0.25% Triton in CSB, briefly rinsed in CSB and incubated with 0.5% glutaraldehyde in CBS for 10 min. The coverslips were then rinsed once rapidly, transferred on to parafilm, and incubated

with 0.5 μM biotinylated phalloidin (Biotin-XX phalloidin, Thermo-Fisher Scientific/Molecular Probes) in PBS with 1% BSA and 0.1% fish gelatin (CWFS gelatin, Aurion) for 1 h, followed by 50 mM NH₄Cl in PBS (15 min in total with a change to fresh solution after 7 min; ref.[39]) and 0.1% fish gelatin in PBS (30 min in total with a change to fresh solution after 15 min). The coverslips were then incubated for 3 h in a humidifying chamber with 10 nm gold-coupled anti-biotin antibodies (Aurion) in 0.1% fish gelatin in PBS and 0.01% BSA-c (Aurion). The coverslips were washed three times 10 min each in gelatin (0.1% in PBS), followed by 1 min in 0.05 % Triton in PBS and four times 1 min washes in PBS. The coverslips were then post-fixed with 1% glutaraldehyde/5 μM phallacidin in CBS for 15 min, transferred to distilled water and processed for osmium/HMDS procedure as above. Samples were imaged by secondary electron detection for SEM and backscatter for gold particle labelling, and combined for image acquisition.

**FESEM colorisation and filament measurements**. Filaments were highlighted using the Brush tool in Adobe Photoshop with 20–50% opacity and pseudo-colours were applied using the Hue/Saturation tool in Photoshop to avoid obscuring the structural details. Filament diameter was measured using the line profiling function in FIJI/ImageJ software on four different FESEM images of melan-a cells at two different resolution (25 K and 40 K). Filaments emanating from melanosomes were measured by racing them individually on FESEM images of melan-a ($n = 6$) and melan-f ($n = 3$) cells. Statistical significance was determined by Mann–Whitney test after evaluating data distribution normality by D'Agostino and Pearson normality test using GraphPad Software.

**Cytoskeleton preparations for rapid freeze/freeze dry/metal replica**. For metal replica microscopy[39], melan-a cells were grown on customised right trapezoid-shaped small glass coverslips for 24 h, and cytoskeletons were extracted with 0.25% Triton in CBS for 45 s. The cytoskeletons were rinsed briefly in CBS, incubated with 0.2 mg/ml myosin S1 (Cytoskeleton Inc., Denver) in PEM buffer (100 mM PIPES-KOH, pH 6.9, 1 mM MgCl₂ and 1 mM EGTA) for 20 min and fixed in with 1% glutaraldehyde in CBS for 15 min. The coverslips were washed twice for 5 min each with distilled water and then plunge-frozen in a liquid nitrogen-cooled 1:4 mixture of isopentane/propane. The samples were transferred to the specimen mount of a freeze fracture unit (Balzers BAF400D) and rotary shadowed at a 45 degree angle with 1.2–1.3 nm tantalum-tungsten or 1.5 nm platinum, followed by respectively 2.5 or 5 nm carbon at 90 degree angle. The replicas were separated from the glass coverslips with 8% hydrofluoric acid, washed into distilled water and picked up onto the surface of formvar-coated copper grids. Samples were examined using a JEOL 1200 EXII transmission electron microscope operating at 80 kV. Images were inverted and processed for increased contrast in Photoshop.

**Metabolic labelling and purification of Fmn2 with myristate analogue YnMyr**. Briefly, HEK293a cells were transfected with plasmids allowing expression of the N-terminus 153 amino acids of FMN2 (with/without the G2A mutation) as an N-terminus fusion to GFP, and incubated with NMTi IMP-1088 (100 nM) and YnMyr (tetradic-13-ynoic acid, 1 μM) as indicated for 24 h. Cells were harvested by trypsinisation and washed with 1× PBS. Pellets were homogenised by sonication in 120 μl lysisbuffer containing 1% (v/v) Triton X-100, 0.1% (w/v) SDS and EDTA-free protease inhibitor cocktail, in PBS (pH 7.4). Homogenates were centrifuged (10,000 × $g$, 4 min, 4 °C), supernatant was collected, protein concentrations were determined. Bio-orthogonal CLICK-ligation of 350 μg protein with capture reagent AzTB were performed as described earlier[74]. Aliquots containing 20 μg protein were kept aside to compare AzTB labelling and abundance of proteins at various stages, namely input, after enrichment with Neutravidin-agarose beads (enriched from 300 μg) and the pull-down supernatant (20 μg). Protein samples were resolved by a standard 12% SDS–PAGE gel running at 90 V, whereafter AzTB-clicked YnMyr-labelled proteins were visualised by fluorescence scanning on a Typhoon Variable Mode Imager 9500 (GE Healthcare). Hereafter, proteins were immobilised on a nitrocellulose membrane using wet blotting (Bio-RAD). Proteins Fmn2-GFP and Fmn2[G2A]-GFP were detected by western blotting with GFP antibody (1:1000, mouse), HSP90 (1:1000, mouse-anti-human Santa Cruz #sc-69703) and ARL1 (1:2000, rabbit-anti-human Proteintech 16012-1-AP), bound subsequently by HRP-conjugated secondary antibodies (1:10,000, Advansta anti-mouse R-05071-500 and anti-rabbit R-05072-500) and detected, using HRP Luminata (Merck) and an ImageQuant LAS4000 (GE Healthcare).

**Immunoprecipitation of Rab27a**. Eight, 14 and 28 × 10 cm dishes of 80% confluent B16-F1 mouse melanoma cells, infected with adenovirus vectors expressing GFP, GFP-Mlph or GFP-hsSPIRE1, and cultured for a 10 days to allow expression of proteins. Cells were then washed with cold PBS, trypsinised to detach from plates, harvested by centrifugation and lyzed in 500 μl of buffer containing 20 mM Tris-HCl, pH 7.5, 50 mM NaCl, 1 mM EGTA, 1% CHAPS, 1× protease inhibitor cocktail (Roche, Mannheim Germany. Product 05 892 970 001) 1 mM GTP. Lysates were centrifuged at 10,000 × $g$ for 15 min and the supernatant was then incubated with goat anti-Rab27a polyclonal antibodies (1 mg; Sicgen, Portugal, Product Ab0023-200) previously immobilised on 50 μl of protein A/G UltraLink Resin (Thermos scientific Rockford IL, Product # 53132). After overnight incubation at 4 °C, the beads were

precipitated by low speed centrifugation, washed with the same buffer and analysed using mass spectrometry to identify co-precipitating proteins.

**Sample preparation and mass spectrometry**. Each IP sample was washed in 200 μL 50 mM triethyl ammonium bicarbonate (TEAB; Sigma, UK, Product #T7408) before centrifugation (3000 × g, 30 s) and careful removal of the supernatant. The samples were then resuspended in 100 μL 50 mM TEAB and were reduced with dithiothreitol (5 mM DTT, 56 °C, 20 min; Sigma, UK, Product #D9779) and alkylated using iodoacetamide (55 mM IAA, room temperature, 15 min in the dark; Sigma, UK, Product #I1149). Tryptic digestion was performed as follows; 1 μg trypsin (Promega, UK, Product #V5111) was added to each sample along with 1 % protease max (20 ng/μL; Promega, UK, Product #V207A) and each sample was incubated at 37 °C in a thermomixer (650 r.p.m.) for 2 h. After 2 h a further 1 μg of trypsin was added and incubation at 37 °C in a thermomixer (650 rpm) continued for 16 h. A further 1 μg of trypsin was added to each sample after 16 h incubation and the samples were incubated at 47 °C in a thermomixer (650 r.p.m.) for 1 h. Samples were concentrated and de-salted using HyperSep C18 spin tips (10–200 μL size; Thermo Scientific, UK, Product #60109-412) using the manufacturers protocol. The samples were concentrated using a vacuum concentrator before resuspension in 5% acetonitrile + 0.1% formic acid (Merck, UK, Product #1.59002.2500). Each sample was analysed on a Sciex TripleTOF 6600 mass spectrometer coupled in line with a eksigent ekspert nanoLC 425 system running in micro flow (5 μL/min) mobile phase B (100% acetonitrile + 0.1% formic acid; Merck, UK, Product #1.59002.2500) over mobile phase A (0.1% formic acid; Merck, UK, Product #1.59013.2500). Samples (4 μl) were injected and trapped onto a YMC Triart C18 pre-column (5 mm, 3 μm, 300 μm ID; mobile phase A; 0.1% formic acid, B; acetonitrile with 0.1 % formic acid) at a flow rate of 10 μL/min. The sample was then eluted off the pre-column onto the YMC Triart C18 analytical column (15 cm, 3 μm, 300 μm ID) in line to a Sciex TripleTOF 6600 Duospray Source using a 50 μm electrode, positive mode +5500 V. Samples were analysed in IDA (Information Dependent Acquisition) mode. The following linear gradient was used: mobile phase B increasing from 3 to 30% over 68 min; 30 to 40% over 5 min, 40 to 80% for column wash and re-equilibration over 2 min (total run time 87 min). IDA acquisition mode was used with a top[30] ion fragmentation (TOFMS m/z 400–1250; product ion 100–1500) followed by 15 s exclusion using rolling collision energy, 250/50 ms accumulation time (TOFMS/product ion; 1.8 s cycle). IDA data were searched using ProteinPilot 5.0.2 (iodoacetamide alkylation, thorough search with emphasis on biological modifications) against the SwissProt mouse database (January 2019) with the relevant human sequence added.

**Statistics and reproducibility**. Unless otherwise stated all experiments were repeated independently at least three times and yielded similar results. In cases where data from one experiment are shown this is representative of the results of all three experiments.

**Reporting summary**. Further information on research design is available in the Nature Research Reporting Summary linked to this article.

## Data availability

The data that support the findings of this study are available from the corresponding author upon reasonable request. The mass spectrometry proteomics data have been deposited to the ProteomeXchange Consortium via PRIDE[75] partner repository with the dataset identifier PXD018083. The Source data underlying Figs. 1c–e, g, 2b, d–f, h, 3m, 4c, 5c, 6b–f, 7a–e and 8c, and Supplementary Figs. 1a–c, f, 3c, 4c, 5c, 6, 8b, 9b and 10c, d are provided as a Source data file. Source data are provided with this paper.

## Code availability

The simulations (Supplementary Fig. 11, Supplementary Movies 1–3) were performed using the Open Source project Cytosim. The simulations can be reproduced using the configuration files attached in the supplementary material, and the source code obtained at https://gitlab.com/f.nedelec/cytosim/-/tags/melanosome. Source data are provided with this paper.

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

## Acknowledgements

We thank Dr. Nick Holliday (University of Nottingham, UK) for vYFP BiFC vectors, Dr. Markus Dettenhofer (CEITEC—Central European Institute of Technology, Brno, Czech republic), Dr. Volney Sheen (Harvard Medical School, Boston, USA) and Dr. Monserrat Bosch (Institut Pasteur, Paris, France) for FMN1pro mutant mice and FMN1-specific antibodies, Prof. Peter Shaw for access to the Axiovert S100 fluorescence microscope, Dr. Andrew Bennett, Niki De Vivo and Monika Owen for help with Q-RT-PCR, Tim Self (confocal microscopy), Dr. Chris Gell (Image analysis), Sue Cooper, Carol Sculthorpe (general; all University of Nottingham, UK), Annette Samol-Wolf (expression vector cloning, University Hospital Regensburg, Germany) for technical assistance, Dr. Anwen Bullen and Dr. Andrew Forge (UCL Ear Institute) for electron microscopy assistance and advice, and Prof. Clare Futter (UCL, Institute of Ophthalmology, UK) for advice on electron microscopy studies. This work was supported by a Medical Research Council New Investigator Award to A.N.H. (grant reference G1100063), Wellcome Trust Grant Awards (grant reference 108429/Z/15/Z to E.V.S. and 204843/Z/16/Z to A.N.H.), a Biotechnology and Biological Sciences Research Council (grant reference BB/F016956/1) and University of Nottingham funded PhD studentship awarded to C.L.R. D.A.B. was funded by Biochemical Society Vacation Studentship. L.M. is funded through the Spanish Ministry of Economy. Industry and Competitiveness (MINECO) [BIO2015-70978-R]. E.K. and T.W. are funded by DFG SPP1464, KE 447/10-2" and DFG KE 447/18-1. F.S. is funded by DFG Research Training Group, GRK 2174 Award. N.A. is supported by a PhD studentship from Taibah University, Medina, Kingdom of Saudi Arabia. W.W.K. and E.W.T. are funded by The Royal Society (Newton International Fellowship grant NF161582 to W.W.K.), European Commission (Marie Sklodowska Curie Individual Fellowship grant 752165 to W.W.K.), and Cancer Research UK (C29637/A20183 to E.W.T.). A.K.M. acknowledges financial support of the John and Lucille van Geest Foundation.

## Author contributions

N.A., C.L.R., T.W., E.L.P., D.A.B., F.S., A.S., J.R., D.A.E., W.W.K., M.B., E.K., A.K.M., F.N. and A.N.H. carried out experiments and analysed data. A.N.H. wrote the manuscript. M.B., E.W.T., A.K.M., F.N. and E.K. contributed to revision of the manuscript. M.C., L.M., P.S.G. and E.V.S. generated the immortal Fmn1-deficient melanocyte cell line melan-f.

## Competing interests

The authors declare no competing interests.
