## [Peer Review File · Nature Communications]

Reviewers' comments:

Reviewer #1 (Remarks to the Author):

Melanosome dispersion in mammalian epidermal melanocytes is known to require a trimeric protein complex composed of Rab27a, its effector melanophilin (Mlph) and an actin-based motor myosin-Va. The authors' group previously showed that dynamic actin tracks are essential for myosin-Va-dependent melanosome dispersion, but their regulatory mechanism is poorly understood in melanocytes. In this manuscript the authors screened actin regulators by a small siRNA library and identified SPIRE1/2 and formin-1 (FMN1) as important actin regulators that mediate melanosome dispersion. Interestingly, the C-terminal domain of SPIRE1/2 shows weak similarity to the SHD, a well-known Rab27a effector domain, and SPIRE1/2 can actually bind to Rab27a in vitro with much lower affinity than Mlph. Moreover, they showed that Myo-SPIRE, an artificial fusion protein between SPIRE1/2 and myosin-Va motor/lever, rescued the melanosome clustering phenotype observed in melan-In cells. Based on these findings, the authors proposed that Rab27a regulates actin-dependent melanosome transport by coordinating the motor protein myosin-Va and track assembly proteins SPIRE1/2 and FMN1.

Overall, the involvement of SPIRE1/2 and FMN1 in actin-dependent melanosome dispersion is well documented and the results obtained are largely convincing. Unfortunately, however, the interaction between Rab27a and SPIRE1/2 was only shown by overexpression of recombinant proteins in HEK293 cells, and hence it is not clear whether Rab27a endogenously interacts with SPIRE1/2 in melanocytes and whether the interaction promotes recruitment of SPIRE1/2 to melanosomes. To strengthen the authors' conclusions, following points need to be addressed before publication.

Specific points:

1. What is lacking in this manuscript is direct evidence for the endogenous Rab27a-SPIRE1/2 interaction in melanocytes. Overexpression study is insufficient to show whether SPIRE functions as a bona fide Rab27a effector in melanocytes. Co-IP experiments should be performed by using specific antibodies (related to Fig. 2B-F). Co-IP data on the Mlph-Rab27a complex is necessary as a positive control. In addition, co-localization between Rab27a and SPIRE1 should be tested at the endogenous protein level (related to Fig. 3).
2. In Fig. 2, GTP/GDP-binding preference should be tested by using wild-type Rab27a protein rather than its GTP/GDP-locked mutants.
3. Because SPIRE has already been shown to bind to Rab11 and Rab3 in other cells, it is important to test its Rab binding specificity in detail. Such data is essential to evaluate whether SPIRE indeed functions as a Rab27a effector rather than a Rab11 or Rab3 effector in melanocytes.
4. SPIRE-knockdown-rescue experiments shown in Fig. 5 are very nice, but one additional point mutant (i.e., Rab27a-binding-defective mutant) should be tested. Because most of the amino acids responsible for Rab27a binding in Mlph are also conserved in SPIRE1/2, it is easy to prepare point mutants that specifically lack Rab27a binding activity. If the authors' hypothesis is true, such mutants should not rescue the melanosome clustering phenotype.
5. The hyper-dispersed phenotype induced by SPIRE2 (Fig. 5) seems to be interesting. Although the authors suggested that low affinity Rab27a binding activity of SPIRE2 is related to this phenotype, this reviewer cannot understand the reason why the weak Rab27a binding ability causes hyper-dispersion of melanosomes. In addition, the authors only evaluated the melanosome distribution area, but it is important to measure actin dynamics and speed of actin-dependent melanosome transport in the presence or absence of SPIRE2. The authors should test this and discuss the possible reason.
6. On page 15, the authors proposed that Rab27a recruits two effector complexes to

melanosomes. Does this mean one Rab27a molecule simultaneously recruits two effectors? Alternatively, are two distinct Rab27a complexes present on the same melanosome? This point needs to be addressed experimentally.

7. The authors previously showed that SPIRE interacts with myosin-V motor. Is it possible that SPIRE1/2 interact melanocyte-type (+exon-F) myosin-Va and they support melanosome dispersion? To test this possibility, SPIRE1/2 should be expressed in melan-In (and melan-ash) cells and pigment area should be evaluated (related to Fig. 4C).

Other minor points:

8. A list of siRNA mini-library of actin filament regulators should be provided as a supplementary Table for general readers.

9. Based on the IF data shown in Fig. 4B, MyoSPIRE1/2 (and Mini-Va) should be tagged with GFP. Where is GFP, N- or C-terminus of MyoSPIRE1/2? Please add GFP in Fig. 4A.

10. On page 24 (1st line in the second paragraph), Pylypenko et al. was cited as author's last name, but not numbers (ref. 36).

Reviewer #2 (Remarks to the Author):

This is a well-done investigation by Alzahofi and colleagues into the mechanism by which melanosomes make long-range actin-dependent motions in melanocytes. It represents a further step in the gradual erosion of a long-held dogma that actin-based movements in animal cells are always short-range. Pointing to the potentially wider interest of their results, they note molecular similarities to actin-based transport in oocytes, and suggest these proteins might play similar roles in a wide range of cell types. The level of detail provided should allow other researchers to reproduce the work, and the use of statistics all appears appropriate. I find the work largely convincing, but as outlined below, there are several areas in which I suggest the authors perform minor revisions before acceptance.

In short, the authors show that dispersal of melanosomes in the cytoplasm depends on the formin FMN1 and the two SPIRE proteins (SPIRE1 and SPIRE2), which are actin-nucleating factors that interact with FMN1. Using a variety of in vitro and in vivo assays, they demonstrate that SPIRE1/2 interact with the key melanosome marker, Rab27, and are likely localized to melanosomes through this interaction. Moreover, they show that FMN1 requires its ability to interact with actin filaments (presumably barbed ends) and with SPIRE, and that full FMN1 function depends on the formin not being stably tethered to the melanosome surface. They tie their observations into a model, whereby SPIRE/FMN1 interactions at the melanosome surface drive assembly of filaments whose barbed ends elongate away from the organelles, providing substrates for Myosin Va to drive melanosome dispersion.

I have four suggestions/criticisms regarding the reported experimentation in the manuscript.

(1) The authors perform several structure/function assays using fusion proteins expressed in melanocytes (Figs 5, 6, 8, and S5) to identify domains of SPIRE1/2 and FMN1/2 that are critical for function. However, interpretation of negative results here (i.e. constructs that fail to rescue functions) depends on knowing that the relevant fusion protein is being expressed to a comparable level as constructs that rescue. The figures show GFP fluorescence, but this does not demonstrate that the fusion protein remains intact. If possible, anti-GFP western blots should be performed to show that at least some of the fusion protein for each construct is intact in the melanocyte systems used.

(2) The authors use SEM to observe melanosome-associated actin filaments (AFs) in Fig.9. The

authors should outline the criteria they used to unambiguously identify actin filaments, as opposed to the possibility that they are viewing IFs or MTs. Arrows indicating the different filament types, or false-coloring, would help make these results more accessible to the reader. Also related to visualization of AFs, are any changes in F-actin organization (based on phalloidin stain) noted on loss of SPIRE1/2 or FMN1 in these cells? The authors should at least attempt to visualize differences by phalloidin stain (or expression of LifeAct or similar). I recognize that if this population of AFs is minor, differences might not be seen by these methods, but the attempt should at least be made.

(3) The authors performed 120 different RNAi knockdowns, and reported on results for four (FMN1, FMN2, SPIRE1, and SPIRE2). I appreciate that the authors might wish to reserve their other knockdown results for future work, but can they at least give some sense of how common it was for a knockdown to result in melanosome clustering? This would provide the reader with some sense of how unique this phenotype is or is not, and how complicated this system might be.

(4) For the quantitative GST pull-downs, the authors should show the data points that contributed to the calculated curves shown in Fig.2F.

Model, would require cortical/dendrite-localized actin filaments.

Minor criticisms/suggestions relating to the text and model:

(1) The authors provide references in the for FMN2 functions in the introduction. For FMN1, they reference its reported role in limb development (which is controversial, based on absence of limb defects in mice bearing *fmn1* mutations eliminating their FH2 domains [Zuniga et al. 2004. *Genes & Development* 18:1553-1564]), but make no mention of other reported functions for FMN1, including formation of linear actin cables and adherens junctions [Kobielak et al. 2003. *Nat Cell Biol* 6(1):21-30] or the blood-testis barrier [Li et al. 2015. *Endocrinology* 156(8): 2969-2983].

(2) Error bars should defined in the figure legends throughout the paper. Also, for Fig.1D, the legend does not state what the arrows are meant to indicate.

(3) Fig.2 legend wording is a little inaccurate when describing panel D. It indicates only data for SPIRE1-expressing cells are shown, but the scatter plot includes SPIRE2-expressing cells.

(4) Minor typo - first sentence of first full paragraph on page 13 should read "To understand better the role of SPIREs..." rather than "To understand better to role of SPIREs...".

(5) The model presented in Fig.7 is very analogous to how FMN2/SPIRE are thought to work in oocytes (as noted by the authors). The oocyte system is thought to also depend on some FMN2 associated with the cortex, providing an immovable anchor site for actin filaments to allow for centripetal movements, e.g. [Schuh M. (2011) An actin-dependent mechanism for long-range vesicle transport. *Nat Cell Biol* 13(12):1431-1436]. Without a similar fixed population of AFs, I would expect that Myosin Va activity would either cause AFs to slide along melanocytes, rather than cause melanocytes to move, OR cause melanocytes to contract in the center of the cell as they pull on the AFs growing from their neighboring melanocytes' surfaces. Do the authors think there is cortex-associated AFs involved in dispersion? Or do they think there is something specific about melanocytes making such filaments unnecessary?

(6) The issue of SPIRE2-dependent hyper-dispersion of melanosomes is curious. The authors suggest it relates to the relatively weak SPIRE2/Rab27 interaction, but it is not clear to me why a weakened interaction would promote hyper-dispersion. I would have expected the opposite. Can the authors expand on this?

(7) Regarding the sequence alignments, could the authors comment on what they mean by "manual confinements" referred to in the Materials and Methods?

Reviewer #3 (Remarks to the Author):

This paper shows that in melanocytes SPIRE1/2 and formin-1 (FMN1) generate actin tracks required for dispersion of melanosomes in melanocytes driven by myosin Va. SPIRE1/2 are recruited to melanosomes by active, GTP-bound Rab27a. Rab27a binding involves C-terminal membrane binding domain of SPIRE1/2 which is related to Rab binding domains of Rab3/Rab27 effectors, including melanophilin that recruits to melanosomes myosin Va. However, SPIRE1/2 interacts with Rab27a on melanosomes with lower affinity than melanophilin. The authors propose a model for dispersion of melanosomes in melanocytes that involves assembly of actin filaments on the melanosome surface by SPIRE1/2 FMN1 and subsequent movement of melanosomes to the barbed ends of nascent actin filaments by myosin Va.

This is an interesting paper that proposes novel mechanism of intracellular transport that involves generation of actin transport tracks on cargo organelles. Such mechanism may have a general importance and explain actin-based transport of organelles in a wide variety of other types of animal cells. Experimental data reveal details of interaction between SPIRE1/2 and Rab27a and the roles of various domains of SPIRE1/2 and FMN1 molecules in nucleation and assembly of actin filaments. However, there are several concerns that should be addressed before the paper can be published.

1. It remains unclear whether recruitment SPIRE1/2 to melanosomes is critical for their motility. Sequence alignment of SPIRE1/2 with melanophilin (Fig. S3) suggests that these proteins interact with Rab27 via similar mechanisms. Therefore SPIRE1/2 and melanophilin should compete for binding to melanosomes. Given that SPIRE1/2 have significantly lower affinity to Rab27a than melanophilin, the level of endogenous melanosome-bound SPIRE1/2 should be very low and therefore it is not clear if it can significantly contribute to nucleation of actin filaments and melanosome transport. The effect of knockdown of SPIRE proteins on cytoplasmic distribution of melanosomes might be explained by their role in organization of actin transport tracks in a melanosome-independent way. Indeed, SPIRE2-KW that lacks melanosome-targeting domain can disperse melanosomes in SPIRE1/2 depleted melan-a cells almost as efficiently as the full-size SPIRE2. Therefore it appears that binding SPIRE proteins to melanosomes is not essential for melanosome transport, and therefore experimental data seem to be inconsistent with the model proposed by the authors.

2. Description of the proposed model for myosin Va/AF dependent melanosome dispersion does not belong to Results and should be moved to Discussion. Two sections of Results that follow description of the model do not add much to the story. The authors make an effort to validate the proposed model by showing that association of FMN1 but not SPIRE with melanosomes should be transient to allow release of actin filaments. However, this aspect of the model is confusing. Released actin filaments lose contact with melanosomes and it is therefore not clear how melanosome-associated myosin Va finds nascent actin filaments located at a distance from the melanosome surface. The next section describes overexpression of FMN2 that should prove that FMN mis-targeting reduces efficiency of melanosome transport. However, these indirect experiments are not convincing. Additional control studies are required to prove that myristoylation targets FMN2 to the plasma membrane as the authors suggest. Therefore either intracellular distribution of FMN2 should be studied in detail or this section should be removed from the manuscript

3. Fig. 1D. Magnification of images is too small to identify melanocytes with overdispersed melanosomes.

Fig. 1G. GFP-FMN1 (shown in green) does not seem to colocalize with melanosomes (shown in magenta).

4. Fig. 9. In the absence of FMN1 short filaments still form around melanosomes. Whether nucleation of actin filaments on melanosomes occurs in cells lacking SPIRE1/2?

5. Images of actin filaments (figure 9) are of poor quality. The authors should provide better quality EM images of melanocytes and prove that the filaments seen in these images are actin filaments by decorating them with myosin S1.

Authors' response to reviewer comments. Author comments in *italicised* text.

Reviewers' comments:

Reviewer #1 (Remarks to the Author):

Melanosome dispersion in mammalian epidermal melanocytes is known to require a trimeric protein complex composed of Rab27a, its effector melanophilin (Mlph) and an actin-based motor myosin-Va. The authors' group previously showed that dynamic actin tracks are essential for myosin-Va-dependent melanosome dispersion, but their regulatory mechanism is poorly understood in melanocytes. In this manuscript the authors screened actin regulators by a small siRNA library and identified SPIRE1/2 and formin-1 (FMN1) as important actin regulators that mediate melanosome dispersion. Interestingly, the C-terminal domain of SPIRE1/2 shows weak similarity to the SHD, a well-known Rab27a effector domain, and SPIRE1/2 can actually bind to Rab27a *in vitro* with much lower affinity than Mlph. Moreover, they showed that Myo-SPIRE, an artificial fusion protein between SPIRE1/2 and myosin-Va motor/lever, rescued the melanosome clustering phenotype observed in melan-In cells. Based on these findings, the authors proposed that Rab27a regulates actin-dependent melanosome transport by coordinating the motor protein myosin-Va and track assembly proteins SPIRE1/2 and FMN1. Overall, the involvement of SPIRE1/2 and FMN1 in actin-dependent melanosome dispersion is well documented and the results obtained are largely convincing. Unfortunately, however, the interaction between Rab27a and SPIRE1/2 was only shown by overexpression of recombinant proteins in HEK293 cells, and hence it is not clear whether Rab27a endogenously interacts with SPIRE1/2 in melanocytes and whether the interaction promotes recruitment of SPIRE1/2 to melanosomes. To strengthen the authors' conclusions, following points need to be addressed before publication.

Specific points:

1. What is lacking in this manuscript is direct evidence for the endogenous Rab27a-SPIRE1/2 interaction in melanocytes. Overexpression study is insufficient to show whether SPIRE functions as a bona fide Rab27a effector in melanocytes. Co-IP experiments should be performed by using specific antibodies (related to Fig. 2B-F). Co-IP data on the Mlph-Rab27a complex is necessary as a positive control. In addition, co-localization between Rab27a and SPIRE1 should be tested at the endogenous protein level (related to Fig. 3).

Unfortunately due to the lack of suitable SPIRE1/2 specific antibodies we have been unable to carry out co-IP or co-localisation experiments to investigate the interaction/localisation of endogenous Rab27a and SPIRE1/2 and localisation of endogenous SPIRE1/2 (as suggested by the reviewer).

Related to this we have been unable to find literature documenting the control Rab27a-Mlph precipitation experiment suggested by the reviewer. The closest published data on the interaction of endogenous Rab27a and Mlph are studies showing that recombinant myosin-Va tail (melanocyte isoform) coated beads were able to precipitate endogenous Mlph from melanocyte lysates (Wu et al., 2002). We have attempted a similar experiment using a bacterially expressed GST-FMN2-FSI protein to precipitate Rab27a and SPIRE1/2 from cell lysates. Although this was successful using lysates from cells expressing recombinant Rab27a and SPIRE1/2 it was unsuccessful using melanocytes lysates probably due to the low levels of

protein expression and transient nature of FMN-FSI:SPIRE-KIND interaction (Montaville et al., 2014). We are happy to share this data with the reviewer on their request.

In response to the comments on the lack of evidence of interaction of endogenous Rab27a:SPIRE1/2 proteins, we argue that localisation and function studies of SPIRE1/2 and Myo-SPIRE1/2 (Figures 7 and 8) get closer to assessment of endogenous SPIRE1/2 protein interaction and localisation and go beyond assessment of the localisation and function of over-expressed recombinant proteins as they show that the activity and localisation of the expressed SPIRE proteins is dependent upon endogenous, not co-expressed, melanosomal Rab27a. For instance, Myo-SPIRE proteins do not function in Rab27a deficient melan-ash cells and SPIRE1/2 proteins do not fully rescue melanosome transport in SPIRE1/2 depleted cells when their interaction with endogenous Rab27a is compromised by truncation or point mutations. To emphasise this point in the manuscript we have revised the text of the results section to more clearly highlight the contribution of endogenous melanosome localised Rab27a in our observations using expressed SPIRE1/2 proteins.

2. In Fig. 2, GTP/GDP-binding preference should be tested by using wild-type Rab27a protein rather than its GTP/GDP-locked mutants.

We have performed the experiment suggested and present this data in a revised version of Figure S6. The results are consistent with those obtained using GTP/GDP-locked mutants i.e. interaction is stronger when Rab27a is in the active conformation.

3. Because SPIRE has already been shown to bind to Rab11 and Rab3 in other cells, it is important to test its Rab binding specificity in detail. Such data is essential to evaluate whether SPIRE indeed functions as a Rab27a effector rather than a Rab11 or Rab3 effector in melanocytes.

We agree that a detailed characterisation of the Rab binding specificity of SPIRE1/2 proteins would provide some interesting data. However, given the size of the Rab family (60 members in human) and the focus of this study on the role SPIRE proteins in melanosome transport we suggest that such work is beyond the scope of this study and should form part of a future study.

Regarding the possibility that SPIRE1/2 could be working with Rab11 and/or Rab3 in melanocytes, we consider this to be unlikely as siRNA depletion of Rab11 and Rab3 isoforms does not affect melanosome distribution in wild-type melanocytes as would be expected if SPIRE1/2 were to be functioning with those Rabs in melanosome dispersion (Hume et al unpublished data) (Ishida et al., 2012). Moreover previous data from our team indicate that Rab11 and SPIRE1/2 do not directly interact (Pylypenko et al., 2016). This further supports the idea that SPIRE1/2 are unlikely to regulate melanosome dispersion by interaction with Rab11.

4. SPIRE-knockdown-rescue experiments shown in Fig. 5 are very nice, but one additional point mutant (i.e., Rab27a-binding-defective mutant) should be tested. Because most of the amino acids responsible for Rab27a binding in Mlph are also conserved in SPIRE1/2, it is easy to prepare point mutants that specifically lack Rab27a binding activity. If the authors' hypothesis is true, such mutants should not rescue the melanosome clustering phenotype.

As suggested by the reviewer we have generated SPIRE1^{E548K} point mutant in which Rab27a interaction is undetectable (Figure S9). Consistent with previously shown SPIRE1/2

knockdown-rescue data for the SPIRE1/2KW truncations (that lack the C-terminus membrane/Rab binding domain) and intact SPIRE2 (that interacts weakly with Rab27a) we saw that expression of the SPIRE1^{E548K} mutant resulted in an increase in the frequency of cells with hyper-dispersed melanosome distribution compared with wild-type SPIRE1. This suggests that interaction with Rab27a and localisation to melanosomes is important for uniform dispersion of melanosomes as seen in wild-type melanocytes. These data are presented in Figures 4 and S9 of the revised manuscript.

5. The hyper-dispersed phenotype induced by SPIRE2 (Fig. 5) seems to be interesting. Although the authors suggested that low affinity Rab27a binding activity of SPIRE2 is related to this phenotype, this reviewer cannot understand the reason why the weak Rab27a binding ability causes hyper-dispersion of melanosomes. In addition, the authors only evaluated the melanosome distribution area, but it is important to measure actin dynamics and speed of actin-dependent melanosome transport in the presence or absence of SPIRE2. The authors should test this and discuss the possible reason.

We agree that this would be a very interesting topic to address. However, given that our functional and gene expression data (Figure S1) indicate a relatively minor role for SPIRE2 versus SPIRE1 in melanosome transport we suggest that in depth studies of SPIRE2 function and AF dynamics in melanocytes are beyond the scope of the current study and should be the subject of future studies.

To reflect the dominant role of SPIRE1 we have re-written the manuscript to present the role of SPIRE1 in transport as the main focus. On the topic of hyper-dispersion we have revised the discussion section to present possible mechanisms by which this might occur. In particular due to its low affinity for interaction with Rab27a we suggest that at low concentration SPIRE2 may interact with other Rab GTPases located on compartments at the cell periphery, thus resulting in assembly of dynamic AFs at the cell periphery and accumulation of melanosomes there. Meanwhile at higher concentrations it may better interact with Rab27a leading to uniform melanosome dispersion by the mechanism proposed in Figure 9.

6. On page 15, the authors proposed that Rab27a recruits two effector complexes to melanosomes. Does this mean one Rab27a molecule simultaneously recruits two effectors? Alternatively, are two distinct Rab27a complexes present on the same melanosome? This point needs to be addressed experimentally.

As suggested by the reviewer we consider the latter possibility i.e. that distinct Rab27a:effector complexes exist on the melanosome membrane, to be the more likely situation. For several reasons we consider it unlikely that Rab27 interacts simultaneously with two different effectors. Firstly, structural studies of 1:1 complexes of Rab27 with effector fragments (Mlph (aka Slac2-a) and synaptotagmin-like protein 2a (Sytl2a aka Slp2a or exophilin-4)) reveal large interaction interfaces (2534 and 1970 Å², equivalent to 1/3 of the total surface area of the Rab27:effector complex). Moreover the interaction interfaces encompass almost entirely the switch and inter-switch regions of Rab27a that are essential for ensuring the specificity of effector binding (Chavas et al., 2008, Kukimoto-Niino et al., 2008). This means that these regions are likely to be obscured and unavailable for interaction with additional effectors. Secondly, as shown here, the Rab27 binding elements of SPIRE1/2 and other effectors e.g. Mlph, are highly related at the level of amino acid similarity, indicating a similar mechanism of interaction i.e. involving extensive contact with Rab27 switch regions. Our results with the SPIRE1^{E548K} mutant support this idea (Figures 4

and S9). As this is likely to be the case we consider that tripartite interaction of Rab27a with 2 effectors (e.g. SPIRE1 and Mlph) simultaneously is highly unlikely. In light of this and given the focus of the paper on defining the mechanism of dynamic actin and myosin-Va in organelle dispersal, rather than the detail of Rab27:effector interaction, we have not directly investigated this point. Instead we have added comments to the discussion of the revised manuscript to explain why we think Rab27a is likely to form 2 different effector complexes at the melanosome membrane.

7. The authors previously showed that SPIRE interacts with myosin-V motor. Is it possible that SPIRE1/2 interact melanocyte-type (+exon-F) myosin-Va and they support melanosome dispersion? To test this possibility, SPIRE1/2 should be expressed in melan-In (and melan-ash) cells and pigment area should be evaluated (related to Fig. 4C).

We have addressed this point by testing the ability of SPIRE1 to rescue melanosome transport defects in Mlph deficient (melan-In) melanocytes. We found that, in contrast to Mlph, SPIRE1 was unable to disperse melanosomes. Thus the significance of the SPIRE1/2:myosin-Va interaction in melanocytes remains to be established. This data is presented in Figure S8 of the revised manuscript.

Other minor points:

8. A list of siRNA mini-library of actin filament regulators should be provided as a supplementary Table for general readers.

This has been added to the revised manuscript in Table S1.

9. Based on the IF data shown in Fig. 4B, MyoSPIRE1/2 (and Mini-Va) should be tagged with GFP. Where is GFP, N- or C-terminus of MyoSPIRE1/2? Please add GFP in Fig. 4A.

The GFP is located at the N-terminus of MyoSPIRE1/2 proteins this is now shown in the revised Figure 8A.

10. On page 24 (1st line in the second paragraph), Pylypenko et al. was cited as author's last name, but not numbers (ref. 36).

This has been corrected in the revised manuscript.

Reviewer #2 (Remarks to the Author):

This is a well-done investigation by Alzahofi and colleagues into the mechanism by which melanosomes make long-range actin-dependent motions in melanocytes. It represents a further step in the gradual erosion of a long-held dogma that actin-based movements in animal cells are always short-range. Pointing to the potentially wider interest of their results, they note molecular similarities to actin-based transport in oocytes, and suggest these proteins might play similar roles in a wide range of cell types. The level of detail provided should allow other researchers to reproduce the work, and the use of statistics all appears appropriate. I find the work largely convincing, but as outlined below, there are several areas in which I suggest the authors perform minor revisions before acceptance.

In short, the authors show that dispersal of melanosomes in the cytoplasm depends on the formin FMN1 and the two SPIRE proteins (SPIRE1 and SPIRE2), which are actin-nucleating

factors that interact with FMN1. Using a variety of in vitro and in vivo assays, they demonstrate that SPIRE1/2 interact with the key melanosome marker, Rab27, and are likely localized to melanosomes through this interaction. Moreover, they show that FMN1 requires its ability to interact with actin filaments (presumably barbed ends) and with SPIRE, and that full FMN1 function depends on the formin not being stably tethered to the melanosome surface. They tie their observations into a model, whereby SPIRE/FMN1 interactions at the melanosome surface drive assembly of filaments whose barbed ends elongate away from the organelles, providing substrates for Myosin Va to drive melanosome dispersion.

I have four suggestions/criticisms regarding the reported experimentation in the manuscript.

(1) The authors perform several structure/function assays using fusion proteins expressed in melanocytes (Figs 5, 6, 8, and S5) to identify domains of SPIRE1/2 and FMN1/2 that are critical for function. However, interpretation of negative results here (i.e. constructs that fail to rescue functions) depends on knowing that the relevant fusion protein is being expressed to a comparable level as constructs that rescue. The figures show GFP fluorescence, but this does not demonstrate that the fusion protein remains intact. If possible, anti-GFP western blots should be performed to show that at least some of the fusion protein for each construct is intact in the melanocyte systems used.

We have carried out the western blots suggested by the reviewer and these have been added to the revised manuscript (Figure S4).

(2) The authors use SEM to observe melanosome-associated actin filaments (AFs) in Fig.9. The authors should outline the criteria they used to unambiguously identify actin filaments, as opposed to the possibility that they are viewing IFs or MTs. Arrows indicating the different filament types, or false-coloring, would help make these results more accessible to the reader. Also related to visualization of AFs, are any changes in F-actin organization (based on phalloidin stain) noted on loss of SPIRE1/2 or FMN1 in these cells? The authors should at least attempt to visualize differences by phalloidin stain (or expression of LifeAct or similar). I recognize that if this population of AFs is minor, differences might not be seen by these methods, but the attempt should at least be made.

To justify the assignment of filaments in SEM images as AFs we have measured the filaments and found them to be of average diameter 8.7 +/- 0.3 nm. This is consistent with the filaments being AFs rather than MTs. This data is included in the results section text of the revised manuscript. Also SEM images show that many filaments are branched indicating that they are unlikely to be MTs. As suggested we have added colours and arrows to a new version of the Figure (Figure 3) presenting SEM data in the revised manuscript to highlight filament architecture.

In response to the reviewer comments about phalloidin staining, we have moved previously shown phalloidin stain images of wild-type and FMN1 deficient melanocytes from the supplementary into the main figures (revised Figure 2). We also present new data showing the difference in AF content between wild-type, FMN1 deficient and SPIRE1/2 depleted melanocytes and the recovery of this AF deficit upon re-expression of FMN1 as seen by phalloidin staining and fluorescence microscopy (revised Figure 2).

(3) The authors performed 120 different RNAi knockdowns, and reported on results for four (FMN1, FMN2, SPIRE1, and SPIRE2). I appreciate that the authors might wish to reserve

their other knockdown results for future work, but can they at least give some sense of how common it was for a knockdown to result in melanosome clustering? This would provide the reader with some sense of how unique this phenotype is or is not, and how complicated this system might be.

A list of the siRNA targets tested in this study has been added to the revised manuscript (Table S1). We have also updated the first section of the results text (page 6) to indicate that in addition to FMN/SPIRE1/2 depletion of 5 other targets resulted in melanosome clustering in >2/5 transfections.

(4) For the quantitative GST pull-downs, the authors should show the data points that contributed to the calculated curves shown in Fig.2F.

We have revised the Figure displaying this data (Figure 6F) so that the curve shows the data points that contributed to the curve.

Model, would require cortical/dendrite-localized actin filaments.

Minor criticisms/suggestions relating to the text and model:

(1) The authors provide references in the for FMN2 functions in the introduction. For FMN1, they reference its reported role in limb development (which is controversial, based on absence of limb defects in mice bearing *fmn1* mutations eliminating their FH2 domains [Zuniga et al. 2004. *Genes & Development* 18:1553-1564]), but make no mention of other reported functions for FMN1, including formation of linear actin cables and adherens junctions [Kobielak et al. 2003. *Nat Cell Biol* 6(1):21-30] or the blood-testis barrier [Li et al. 2015. *Endocrinology* 156(8): 2969-2983].

We have updated the referencing in the introduction section of the revised manuscript dealing with FMN1 function to better support the role of FMN1 in limb development. In particular we now cite (Zhou et al., 2009) who directly tested the role of FMN1 in limb development and found that disruption of FMN1 expression resulted in oligodactylyism in mice. We have not cited the Kobielak 2003 paper (Kobielak et al., 2004) or mentioned the link between FMN1 and adherens junctions as results presented in a more recent study (Dettenhofer et al., 2008) indicate that the antibody used in the Kobielak 2003 study to detect FMN1 by immunofluorescence and link it spatially to adherens junctions detects actin rather than FMN1. Additionally localisation studies by Dettenhofer et al 2008, using cells derived from a GFP-FMN1 knock-in mouse, did not support the association of FMN1 with adherens junctions (Dettenhofer et al., 2008). Finally, as advised by the reviewer we have revised the introduction text to include mention of other reported functions on FMN1 in neurogenesis and spermatogenesis (Li et al., 2015, Simon-Areces et al., 2011).

(2) Error bars should defined in the figure legends throughout the paper. Also, for Fig.1D, the legend does not state what the arrows are meant to indicate.

The bars in the scatter plots shown throughout the paper indicate the mean and 25th and 75th percentile of data. This is now indicated in the figure legends in the revised manuscript.

The legend of Figure 1 has been revised to confirm that the arrows in Figure 1D (Figure 1G in revised manuscript) highlight the position of melanocytes in which melanosomes have hyper-dispersed distribution.

(3) Fig.2 legend wording is a little inaccurate when describing panel D. It indicates only data for SPIRE1-expressing cells are shown, but the scatter plot includes SPIRE2-expressing cells.

The legend of revised Figure 6D has been updated to indicate that the scatter plot reports data from both SPIRE1 and SPIRE2 expressing cells.

(4) Minor typo - first sentence of first full paragraph on page 13 should read "To understand better the role of SPIREs..." rather than "To understand better to role of SPIREs...".

This typo has been removed during the revision of the manuscript.

(5) The model presented in Fig.7 is very analogous to how FMN2/SPIRE are thought to work in oocytes (as noted by the authors). The oocyte system is thought to also depend on some FMN2 associated with the cortex, providing an immovable anchor site for actin filaments to allow for centripetal movements, e.g. [Schuh M. (2011) An actin-dependent mechanism for long-range vesicle transport. Nat Cell Biol 13(12):1431-1436]. Without a similar fixed population of AFs, I would expect that Myosin Va activity would either cause AFs to slide along melanocytes, rather than cause melanocytes to move, OR cause melanocytes to contract in the center of the cell as they pull on the AFs growing from their neighboring melanocytes' surfaces. Do the authors think there is cortex-associated AFs involved in dispersion? Or do they think there is something specific about melanocytes making such filaments unnecessary?

We have updated the model (revised Figure 9) to suggest that AFs generated by FMN/SPIRE are anchored at the melanosome membrane as seen in SEM images (Figures 3 and S2) thereby allowing the power-stroke of myosin-Va to disperse melanosomes from one another.

(6) The issue of SPIRE2-dependent hyper-dispersion of melanosomes is curious. The authors suggest it relates to the relatively weak SPIRE2/Rab27 interaction, but it is not clear to me why a weakened interaction would promote hyper-dispersion. I would have expected the opposite. Can the authors expand on this?

We have revised the discussion section of the manuscript (page 20) to address this point. In particular we suggest that SPIRE2 may interact with other Rabs associated with compartments located at the cell periphery. Thus resulting in assembly of AFs at the cell periphery and accumulation of melanosomes there.

(7) Regarding the sequence alignments, could the authors comment on what they mean by "manual confinements" referred to in the Materials and Methods?

We have revised the sequence alignment (Figure S5A) so that this shows the results of Clustal Omega analysis of the shown sequences with no other modification or manual confinement. In accord with this we have revised the description of the construction of the alignment in the Experimental procedures section (see page 24-25).

Reviewer #3 (Remarks to the Author):

This paper shows that in melanocytes SPIRE1/2 and formin-1 (FMN1) generate actin tracks required for dispersion of melanosomes in melanocytes driven by myosin Va. SPIRE1/2 are recruited to melanosomes by active, GTP-bound Rab27a. Rab27a binding involves C-

terminal membrane binding domain of SPIRE1/2 which is related to Rab binding domains of Rab3/Rab27 effectors, including melanophilin that recruits to melanosomes myosin Va. However, SPIRE1/2 interacts with Rab27a on melanosomes with lower affinity than melanophilin. The authors propose a model for dispersion of melanosomes in melanocytes that involves assembly of actin filaments on the melanosome surface by SPIRE1/2 FMN1 and subsequent movement of melanosomes to the barbed ends of nascent actin filaments by myosin Va.

This is an interesting paper that proposes novel mechanism of intracellular transport that involves generation of actin transport tracks on cargo organelles. Such mechanism may have a general importance and explain actin-based transport of organelles in a wide variety of other types of animal cells. Experimental data reveal details of interaction between SPIRE1/2 and Rab27a and the roles of various domains of SPIRE1/2 and FMN1 molecules in nucleation and assembly of actin filaments. However, there are several concerns that should be addressed before the paper can be published.

1. It remains unclear whether recruitment SPIRE1/2 to melanosomes is critical for their motility. Sequence alignment of SPIRE1/2 with melanophilin (Fig. S3) suggests that these proteins interact with Rab27 via similar mechanisms. Therefore SPIRE1/2 and melanophilin should compete for binding to melanosomes. Given that SPIRE1/2 have significantly lower affinity to Rab27a than melanophilin, the level of endogenous melanosome-bound SPIRE1/2 should be very low and therefore it is not clear if it can significantly contribute to nucleation of actin filaments and melanosome transport. The effect of knockdown of SPIRE proteins on cytoplasmic distribution of melanosomes might be explained by their role in organization of actin transport tracks in a melanosome-independent way. Indeed, SPIRE2-KW that lacks melanosome-targeting domain can disperse melanosomes in SPIRE1/2 depleted melan-a cells almost as efficiently as the full-size SPIRE2. Therefore it appears that binding SPIRE proteins to melanosomes is not essential for melanosome transport, and therefore experimental data seem to be inconsistent with the model proposed by the authors.

We agree with the reviewer that our data indicate that SPIRE1/2:Rab27a interaction is not absolutely essential for motility/dispersion of melanosomes. However, our data showing that Rab27a interaction deficient SPIRE1/2-KW truncations and intact SPIRE2, indicate that this interaction is required for efficient distribution of melanosomes into the uniform cytoplasmic dispersion pattern seen in wild-type melanocytes. [We also observe in many experiments that a higher expression level of SPIRE2 and SPIRE1/2-KW proteins is required to drive melanosome dispersion relative to SPIRE1.] These observations, together with the low expression level of endogenous SPIRE2 compared with SPIRE1 in melanocytes, indicates that SPIRE:Rab27a interaction is physiologically required for dispersion.

To directly investigate this we have generated the Rab27a/melanosome interaction deficient SPIRE1^{E548K} mutant and tested its activity in the SPIRE1/2 knockdown-rescue assay. Consistent with the importance of SPIRE1/2:Rab27a interaction in promoting uniform (and not hyper-) dispersal of melanosomes we found that expression of the SPIRE1^{E548K} mutant caused hyper-dispersion of melanosomes. These data are presented in Figures 4 and S9 of the revised manuscript. Thus we conclude that SPIRE1/2:Rab27a interaction is important for the physiological uniform dispersal pattern of melanosomes seen in wild-type melanocytes.

Regarding the possibility that the low affinity of SPIRE1/2:Rab27a compared with Mlph:Rab27a interaction might prevent SPIRE1/2 from associating with melanosomes we

present Q-RT-PCR data showing that SPIRE1 is expressed ~five-fold higher than Mlph in melanocytes (Figure S1F). Based on this we suggest that the relatively high amount of SPIRE1 versus Mlph may compensate for the relatively low affinity of SPIRE1/2:Rab27a versus Mlph:Rab27a interaction and allow SPIRE1 to effectively compete with Mlph and association with melanosome associated Rab27a. This possibility is supported by functional data showing that disruption of SPIRE1/2:Rab27a interaction blocks the ability of SPIRE1/2 uniformly disperse melanosomes.

2. Description of the proposed model for myosin Va/AF dependent melanosome dispersion does not belong to Results and should be moved to Discussion. Two sections of Results that follow description of the model do not add much to the story. The authors make an effort to validate the proposed model by showing that association of FMN1 but not SPIRE with melanosomes should be transient to allow release of actin filaments. However, this aspect of the model is confusing. Released actin filaments lose contact with melanosomes and it is therefore not clear how melanosome-associated myosin Va finds nascent actin filaments located at a distance from the melanosome surface. The next section describes overexpression of FMN2 that should prove that FMN mis-targeting reduces efficiency of melanosome transport. However, these indirect experiments are not convincing. Additional control studies are required to prove that myristoylation targets FMN2 to the plasma membrane as the authors suggest.

Therefore either intracellular distribution of FMN2 should be studied in detail or this section should be removed from the manuscript

Regarding the positioning of the model in the manuscript, we have re-written the results and discussion so that the description of the model is now located in the discussion section of the revised manuscript rather than results section (as previously) and the model is now presented as the final figure (Figure 9).

Regarding the association of SPIRE/FMN with melanosomes and the association of AFs with the melanosome, we have revised the model, based on SEM data showing that AFs emanate from and often bridge adjacent melanosomes, to show that AFs are anchored at the melanosome membrane.

Regarding the myristoylation and membrane localisation of FMN2, to support this we cite previous work (Lian et al., 2016) showing that myristoylation targets FMN2 to cell membranes and confirm that FMN2 is myristoylated by presenting the results of new myristate analogue labelling experiments in the revised manuscript (Figure S10).

3. Fig. 1D. Magnification of images is too small to identify melanocytes with overdispersed melanosomes.

Fig. 1G. GFP-FMN1 (shown in green) does not seem to colocalize with melanosomes (shown in magenta).

Regarding Figure 1D, we have added an inset image in this figure (now Figure 1G) and added red asterisks to highlight cells with over or hyper-dispersed melanosomes. Similar images obtained using a confocal microscope are presented in Figure S1G.

Regarding Figure original 1G, we confirm that GFP-FMN1 does not clearly associate with melanosomes. This is consistent with our model suggesting that FMN1 only transiently associates with melanosomes via interaction with SPIRE1/2 (Figure 9).

4. Fig. 9. In the absence of FMN1 short filaments still form around melanosomes. Whether nucleation of actin filaments on melanosomes occurs in cells lacking SPIRE1/2?

We have investigated this by using FESEM to examine melanosome associated AF structure in melanocytes siRNA depleted of SPIRE1/2. The data are presented in Figure S2 and show that in these cells, as for those lacking FMN1, melanosomes are decorated by short AFs indicating that SPIRE/FMN collaborate to elongate these AFs into the network of inter-melanosome filaments seen in wild-type cells.

5. Images of actin filaments (figure 9) are of poor quality. The authors should provide better quality EM images of melanocytes and prove that the filaments seen in these images are actin filaments by decorating them with myosin S1.

We have revised the images showing AFs in melanocytes (Figures 2 and S2). We hope that the reviewer finds the quality of these images acceptable. To confirm that the filaments are AFs and not microtubules we have measured and found the mean diameter of filaments is 8.7 +/- 0.3 nm. This is consistent with the diameter of AFs ~9 nm and, together with the fact that filaments are sometimes branched, supports the idea that they are AFs.

** See Nature Research's author and referees' website at www.nature.com/authors for information about policies, services and author benefits

This email has been sent through the Springer Nature Tracking System NY-610A-NPG&MTS

Confidentiality Statement:

This e-mail is confidential and subject to copyright. Any unauthorised use or disclosure of its contents is prohibited. If you have received this email in error please notify our Manuscript Tracking System Helpdesk team at <http://platformsupport.nature.com>.

Details of the confidentiality and pre-publicity policy may be found here <http://www.nature.com/authors/policies/confidentiality.html>

Privacy Policy | Update Profile

CHAVAS, L. M., IHARA, K., KAWASAKI, M., TORII, S., UEJIMA, T., KATO, R., IZUMI, T. & WAKATSUKI, S. 2008. Elucidation of Rab27 recruitment by its effectors: structure of Rab27a bound to Exophilin4/Slp2-a. *Structure*, 16, 1468-77.
DETTENHOFER, M., ZHOU, F. & LEDER, P. 2008. Formin 1-isoform IV deficient cells exhibit defects in cell spreading and focal adhesion formation. *PLoS One*, 3, e2497.
ISHIDA, M., OHBAYASHI, N., MARUTA, Y., EBATA, Y. & FUKUDA, M. 2012. Functional involvement of Rab1A in microtubule-dependent anterograde melanosome transport in melanocytes. *J Cell Sci*, 125, 5177-87.

- KOBIELAK, A., PASOLLI, H. A. & FUCHS, E. 2004. Mammalian formin-1 participates in adherens junctions and polymerization of linear actin cables. *Nat Cell Biol*, 6, 21-30.
- KUKIMOTO-NIINO, M., SAKAMOTO, A., KANNO, E., HANAWA-SUETSUGU, K., TERADA, T., SHIROUZU, M., FUKUDA, M. & YOKOYAMA, S. 2008. Structural basis for the exclusive specificity of Slac2-a/melanophilin for the Rab27 GTPases. *Structure*, 16, 1478-90.
- LI, N., MRUK, D. D., WONG, C. K., HAN, D., LEE, W. M. & CHENG, C. Y. 2015. Formin 1 Regulates Ectoplasmic Specialization in the Rat Testis Through Its Actin Nucleation and Bundling Activity. *Endocrinology*, 156, 2969-83.
- LIAN, G., DETTENHOFER, M., LU, J., DOWNING, M., CHENN, A., WONG, T. & SHEEN, V. 2016. Filamin A- and formin 2-dependent endocytosis regulates proliferation via the canonical Wnt pathway. *Development*, 143, 4509-4520.
- MONTAVILLE, P., JEGOU, A., PERNIER, J., COMPPE, C., GUICHARD, B., MOGESSIE, B., SCHUH, M., ROMET-LEMONNE, G. & CARLIER, M. F. 2014. Spire and Formin 2 synergize and antagonize in regulating actin assembly in meiosis by a ping-pong mechanism. *PLoS Biol*, 12, e1001795.
- PYLYPENKO, O., WELZ, T., TITTEL, J., KOLLMAR, M., CHARDON, F., MALHERBE, G., WEISS, S., MICHEL, C. I., SAMOL-WOLF, A., GRASSKAMP, A. T., HUME, A., GOUD, B., BARON, B., ENGLAND, P., TITUS, M. A., SCHWILLE, P., WEIDEMANN, T., HOUDUSSE, A. & KERKHOFF, E. 2016. Coordinated recruitment of Spir actin nucleators and myosin V motors to Rab11 vesicle membranes. *Elife*, 5.
- SIMON-ARECES, J., DOPAZO, A., DETTENHOFER, M., RODRIGUEZ-TEBAR, A., GARCIA-SEGURA, L. M. & AREVALO, M. A. 2011. Formin1 mediates the induction of dendritogenesis and synaptogenesis by neurogenin3 in mouse hippocampal neurons. *PLoS One*, 6, e21825.
- WU, X. S., RAO, K., ZHANG, H., WANG, F., SELLERS, J. R., MATESIC, L. E., COPELAND, N. G., JENKINS, N. A. & HAMMER, J. A., 3RD 2002. Identification of an organelle receptor for myosin-Va. *Nat Cell Biol*, 4, 271-8.
- ZHOU, F., LEDER, P., ZUNIGA, A. & DETTENHOFER, M. 2009. Formin1 disruption confers oligodactylism and alters Bmp signaling. *Hum Mol Genet*, 18, 2472-82.

Reviewers' comments:

Reviewer #1 (Remarks to the Author):

In the revised manuscript the authors addressed some of the concerns raised by this reviewer, but the major issue about the endogenous interaction between SPIRE1 and Rab27a in melanocytes has not been settled yet. I have checked the literature on the Rab27a-Mlph interaction studies myself and found several publications showing co-IP data on the endogenous Rab27a-Mlph interaction (e.g., J Biol Chem 2002;277:12432-6; Nat Cell Biol 2004;6:1195-203). Once the endogenous interaction has been shown, I also think that the interaction can be evaluated by using recombinant proteins in subsequent studies. However, because this is the first study to identify SPIRE1 as a Rab27a-binding protein, I believe that their interaction at the endogenous protein level is crucial. If the suitable antibodies for co-IP experiments are unavailable, the authors should try other approaches such as chemical cross-linking. Alternatively, the authors should at least test the interaction between endogenous Rab27a and tagged SPIRE1 (or tagged Rab27a and endogenous SPIRE1) in "melanocytes". Without such data, the authors should not claim that SPIRE1 indeed functions as a Rab27a effector in melanocytes. Aside from this point, following concerns also need to be addressed.

Other points:

1. New Fig. S8 data that shows SPIRE1 failed to rescue melanosome clustering in Mlph-deficient melan-In cells is surprising, but it is a very important data. For general readers, the authors should clearly describe this result in detail in the main text: "SPIRE1 itself does not mediate actin-based melanosome transport, even though SPIRE1 can interact with Rab27a and myosin-Va like Mlph". Otherwise, myo-SPIRE data shown in Fig. 8 mislead the general readers.

2. SPIRE2 is not expressed well in melanocytes (Fig. S1F) and not associated with melanosomes (Fig. 7C) and it hardly interacts with Rab27a even in vitro (Fig. 6B), SPIRE2 is most unlikely to function as a Rab27a effector in melanocytes. Thus, the authors should specify SPIRE1 as a Rab27a effector throughout the text including abstract.

3. A hyper-dispersion phenotype of Rab27a-binding-deficient mutant of SPIRE1 is very interesting (Fig. 4). Does it mean that Rab27a is a negative regulator of the KW domain that intrinsically has an ability to induce hyper-dispersion of melanosomes? If so, some discussions would be helpful for the general readers.

Reviewer #2 (Remarks to the Author):

The authors have largely addressed my concerns from the previous review, with one exception. I had previously emphasized the importance of documenting predicted perturbations of actin organization by a method alternative to FESEM, such as staining with fluorescent phalloidin. The authors have moved their phalloidin stained FMN1-deficient cells, but not for SPIRE1/2-depleted melanocytes, despite indicating in the rebuttal that they "present new data showing the difference in AF content between wild-type, FMND1 deficient and SPIRE1/2 depeleted melanocytes ...". As I indicated in my review, a major recommendation remains that this staining should at least be attempted on SPIRE1/2 depleted melanocytes.

Otherwise, my only other notes are a few additional mistakes/mistypes that may have cropped into this revision:

On page 7, one callout for Figure 1 appears incorect, stating panel F depicts mean pigment area, whereas it is panel E.

On page 8, the first callout for Figure 2 seems like it should be for Figure 2A-C rather than Figure 2

A-B.

The Figure 2 legend misstates that what is shown is, "AF abundance (E) in melanocytes in the presence and absence of latrunculin-A." Only melanosome dispersion is shown in the presence/absence of latrunculin-A. AF abundance is shown only in absence.

The Figure 4 legend discusses asterisks for panel C, but none are shown.

In Figure 6F, the label for the graph's Y-axis appears to be incomplete.

The Figure S4 legend does not indicate the significance of the asterisks that appear in each panel.

In their rebuttal, the authors note that regarding FMN1 function in limb formation, they now cite Zhou et al, 2009. They do so in the Discussion, but this reference should also be cited in the Introduction, where FMN1 in vivo functions are discussed. Also, one reference cited in that portion of the Intro (reference 23) appears only partial (no date indicated), and not necessarily appropriate for in vivo functions.

With correction of these, I would strongly recommend publication of this manuscript.

Reviewer #3 (Remarks to the Author):

Revised paper is much improved and should be published in Nature Communications. However, there are a few points that still need to be addressed before publication.

1. The authors add data on measurements of diameter of filaments associated with melanosomes and find that it is consistent with diameter of actin filaments. However, they do not improve quality of EM images, which remains poor and therefore new data are not convincing. The authors should obtain better images of actin filaments preferably decorated with myosin S1 or remove this section from the paper.

2. In revised paper, the authors describe in more detail experiments that involves treatment cells with Latrunculin (page 8, and new Fig. 2). They find that disruption of actin filaments with Latrunculin reduces dispersion of melanosomes in control melan-a cells but not in melan-f- or SPIR1/2-depleted melanocytes. The authors conclude that FMN1 (and SPIRE1/2) assemble dynamic actin filaments that support myosin-Va-dependent melanosome dispersion. However, in melan-f- or SPIR1/2-depleted melanocytes melanosomes are already clustered in the cell center and it is not surprising that Latrunculin treatment does not change their distribution. Decrease in degree of melanosome dispersion in control melan-a cells indeed suggests that distribution of melanosomes depends on dynamics of actin filaments. However, similarities in melanosome distribution (clustering in the cell center) in Latrunculin-treated melan-a and melan-f- or SPIR1/2-depleted cells does not prove that FMN1 or SPIR proteins are involved in assembly of dynamic actin filaments.

Minor points:

Legend to Fig. 1G says that arrows indicate cells with hyperdispersed melanosomes. However, these cells are marked with asterisks.

Legend to Fig. 2 is confusing. The legend says that "melan-a and melan-f cells were ... transfected with siRNA indicated..." but images presented here do not seem to show transfection of melan-f cells with siRNA. The authors should revise the legend and describe clearly images shown in Fig. 2.

Reviewers' comments:

Reviewer #1 (Remarks to the Author):

In the revised manuscript the authors addressed some of the concerns raised by this reviewer, but the major issue about the endogenous interaction between SPIRE1 and Rab27a in melanocytes has not been settled yet. I have checked the literature on the Rab27a-Mlph interaction studies myself and found several publications showing co-IP data on the endogenous Rab27a-Mlph interaction (e.g., J Biol Chem 2002;277:12432-6; Nat Cell Biol 2004;6:1195-203). Once the endogenous interaction has been shown, I also think that the interaction can be evaluated by using recombinant proteins in subsequent studies. However, because this is the first study to identify SPIRE1 as a Rab27a-binding protein, I believe that their interaction at the endogenous protein level is crucial. If the suitable antibodies for co-IP experiments are unavailable, the authors should try other approaches such as chemical cross-linking. Alternatively, the authors should at least test the interaction between endogenous

Rab27a and tagged SPIRE1 (or tagged Rab27a and endogenous SPIRE1) in "melanocytes". Without such data, the authors should not claim that SPIRE1 indeed functions as a Rab27a effector in melanocytes. Aside from this point, following concerns also need to be addressed.

Unfortunately, antibodies that reliably detect the endogenous levels of SPIRE1 in melanocytes by western blotting are unavailable. Therefore, to address the question of SPIRE1 interaction with endogenous Rab27a we transiently expressed GFP-SPIRE1 in melanocytes, immunoprecipitated (IP) Rab27a and then used mass spectrometry to search for SPIRE1-derived peptides in the IP. We identified 13 unique Rab27a peptides and 18 unique SPIRE1 peptides in the IP (Figure S7, Tables S2 and S3, see yellow highlighted text on page 12 and methods described on pages 35-36). This, added to many other pieces of data presented in the manuscript (see below), supports the idea that SPIRE1 interacts with endogenous Rab27a and that SPIRE1 functions as a Rab27a effector in melanocytes.

Other data supporting interaction of SPIRE1 with endogenous Rab27a in melanocytes.

- 1) *SPIRE1 localisation to melanosomes is dependent upon endogenous Rab27a (Figure 7).*
- 2) *Rescue of melanosome dispersion by MyoSPIRE1 is dependent upon endogenous Rab27a (Figure 8).*
- 3) *SPIRE1 mutants that do not interact with Rab27a do not rescue uniform melanosome dispersion in SPIRE1/2 depleted melanocytes, as seen for wild-type SPIRE1 (Figure 4, S4).*
- 4) *SPIRE2, whose affinity of interaction with Rab27a is lower than SPIRE1, is less efficient in associating with melanosomes and rescuing uniform melanosome dispersion in SPIRE1/2 depleted melanocytes compared with SPIRE1 (Figure 1, 4, S3, 7).*

Related to this we also cite a new proximity proteomic study that detected interaction between endogenous SPIRE1 and Rab27a in HUVEC endothelial cells. This observation further supports the hypothesis that Rab27a and SPIRE1 interact in mammalian cells (see yellow highlighted text on page 12).

Other points:

1. New Fig. S8 data that shows SPIRE1 failed to rescue melanosome clustering in Mlph-deficient melan-In cells is surprising, but it is a very important data. For general readers, the authors should clearly describe this result in detail in the main text: "SPIRE1 itself does not mediate actin-based melanosome transport, even though SPIRE1 can interact with Rab27a and myosin-Va like Mlph". Otherwise, myo-SPIRE data shown in Fig. 8 mislead the general readers.

We have revised the section of the results text describing this observation (page 14) to make it clearer (see below). We also provide some discussion of the possible reasons for the lack of rescue as follows (see yellow highlighted text on page 15).

Previous text: In melan-ln we found that both myoSPIRE proteins, like minimyosin but not Rab27a or SPIRE-1, dispersed melanosomes compared with GFP alone'

Revised text: 'Interestingly SPIRE1 did not rescue melanosome transport, even though it can interact with Rab27a and myosin-Va like Mlph (Figure S9; mean pigment area (% total); GFP = 24.8 +/- 7.91, SPIRE-1 = 18.5 +/- 3.95 and Mlph = 75.4 +/- 18.8). One possibility is that the affinity of SPIRE1:myosin-Va/Rab27a interactions may be too low to recruit sufficient active myosin-Va to melanosomes to drive their dispersal. Consistent with this our data reveal the low affinity of Rab27a:SPIRE1/2 interaction relative to Rab27a:Mlph (Figure 6F). Meanwhile previous studies showed that SPIRE-2 interacts with myosin-Va-GTD (0.9 +/- 0.11 μ M) with lower affinity compared with Mlph (0.5 μ M)^{38,46}.'

2. SPIRE2 is not expressed well in melanocytes (Fig. S1F) and not associated with melanosomes (Fig. 7C) and it hardly interacts with Rab27a even in vitro (Fig. 6B), SPIRE2 is most unlikely to function as a Rab27a effector in melanocytes. Thus, the authors should specify SPIRE1 as a Rab27a effector throughout the text including abstract.

We agree with the reviewer to the extent that SPIRE1 appears to be dominant over SPIRE2 as a Rab27a effector in melanocytes. However, several pieces of data indicate that we cannot exclude the possibility that SPIRE2 also works with Rab27a in melanocytes. Firstly, knockdown of SPIRE1 alone causes melanosome clustering in a lower proportion of melanocytes compared with double SPIRE1/2 depletion (Figure S1D), indicating that SPIRE2 can contribute to melanosome dispersal in the absence of SPIRE1. Secondly, expression of SPIRE2 can drive uniform melanosome dispersal as well as hyper-dispersal in melanocytes depleted of endogenous SPIRE1/2. This type of dispersion is more obvious in cells expressing higher levels of SPIRE2 indicating that it functions at lower efficiency compared with SPIRE1 (Figure 1H and S3C). This is consistent with the results of in vitro pull-down assays showing that SPIRE2 interacts less well with Rab27a compared with SPIRE1 (Figure 6B). Thirdly, the ability of myoSPIRE2 to rescue melanosome clustering in melan-ln (Mlph -/-, in which Rab27a is present on the melanosome membrane), but not melan-ash (Rab27a -/-), cells, indicates that in intact cells SPIRE2 can interact with endogenous Rab27a (Figure 8B-C). Thus, we do not consider that it is entirely accurate to refer solely to SPIRE1 as a Rab27a effector. Nevertheless, we have revised the Abstract and Introduction sections of the manuscript, adding comments similar to those previously present in the Results and Discussion sections, highlighting the dominant role of SPIRE1 over SPIRE2 as a Rab27a effector (see below).

Abstract: 'Here we show that the SPIRE-type actin nucleators (predominantly SPIRE1) are Rab27a effectors that co-operate with formin-1 to generate actin tracks required for myosin-Va-dependent transport in melanocytes.' (see yellow highlighted text).

Introduction: (closing paragraph page 5, yellow highlighted text) 'Here we present evidence that the myosin-Va mediated melanosome transport/dispersion in melanocytes is dependent upon AF assembly activities of FMN1 and SPIRE1/2, and that SPIRE1/2 (predominantly SPIRE1) can be recruited to melanosomes by Rab27a.'

3. A hyper-dispersion phenotype of Rab27a-binding-deficient mutant of SPIRE1 is very interesting (Fig. 4). Does it mean that Rab27a is a negative regulator of the KW domain that intrinsically has an ability to induce hyper-dispersion of melanosomes? If so, some discussions would be helpful for the general readers.

We agree that this is an interesting observation and that Rab27a interaction may serve to regulate the activity of the KW. To include this idea we have revised the discussion text (page 21, see yellow highlighted text).

'This suggests that interaction with Rab27a is important in ensuring uniform dispersion as seen in wild-type melanocytes by toning down and localising the hyper-dispersive activity of the KW to melanosomes.'

Reviewer #2 (Remarks to the Author):

The authors have largely addressed my concerns from the previous review, with one exception. I had previously emphasized the importance of documenting predicted perturbations of actin organization by a method alternative to FESEM, such as staining with fluorescent phalloidin. The authors have moved their phalloidin stained FMN1-deficient cells, but not for SPIRE1/2-depleted melanocytes, despite indicating in the rebuttal that they "present new data showing the difference in AF content between wild-type, FMND1 deficient and SPIRE1/2 depleted melanocytes ...". As I indicated in my review, a major recommendation remains that this staining should at least be attempted on SPIRE1/2 depleted melanocytes.

We have added new data and show this in a revised Figure 2 showing that the amount of melanosome associated F-actin in SPIRE1/2 double depleted melan-a cells is significantly lower than in control NT siRNA transfected melan-a cells (Figure 2C and F). Moreover, we show that the remaining F-actin in SPIRE1/2 double depleted cells is resistant to latrunculin-A. This supports the hypothesis that SPIRE1/2 (along with FMN1) are important for the assembly of dynamic F-actin that is essential for melanosome dispersal. We described these results on page 8 in the first block of yellow highlighted text.

Otherwise, my only other notes are a few additional mistakes/mistypes that may have cropped into this revision:

On page 7, one callout for Figure 1 appears incorrect, stating panel F depicts mean pigment area, whereas it is panel E.

We have corrected this in the revised the manuscript (see yellow highlighted text on page 7).

On page 8, the first callout for Figure 2 seems like it should be for Figure 2A-C rather than Figure 2 A-B.

We have revised this section in response to comment 1 (see above) and corrected this in the revision (see first block of yellow highlighted text on page 8).

The Figure 2 legend misstates that what is shown is, "AF abundance (E) in melanocytes in the presence and absence of latrunculin-A." Only melanosome dispersion is shown in the presence/absence of latrunculin-A. AF abundance is shown only in absence.

We have corrected this in the revised the manuscript and actin filament abundance in melan-a NT and SPIRE1/2 siRNA transfected and melan-f is shown in Figure 2E-F. (See revised figure and legend yellow highlighted text on page 45).

The Figure 4 legend discusses asterisks for panel C, but none are shown.

We have removed this section from the legend of figure 4.

In Figure 6F, the label for the graph's Y-axis appears to be incomplete.

We have reformatted Figure 6F so that the Y-axis is complete.

Th Figure S4 legend does not indicate the significance of the asterisks that appear in each panel.

We have clarified the legend of Figure S4 to address this point as follows. 'Asterisks indicate the bands that are likely to correspond to each full-length GFP fusion protein based on their primary structure (A and B to the left of the band and C to the right of the band).' (See yellow highlighted text on page 2 of supplementary information).

In their rebuttal, the authors note that regarding FMN1 function in limb formation, they now cite Zhou et al, 2009. They do so in the Discussion, but this reference should also be cited in the Introduction, where FMN1 in vivo functions are discussed. Also, one reference cited in that portion of the Intro (reference 23) appears only partial (no date indicated), and not necessarily appropriate for in vivo functions.

We have revised the introduction and now cite Zhou et al 2009 in that section (new citation 25). We have removed previous reference 23 (Dettenhofer et al 2008) in accord with the reviewer's suggestion (see yellow highlighted text on page 4).

With correction of these, I would strongly recommend publication of this manuscript.

Reviewer #3 (Remarks to the Author):

Revised paper is much improved and should be published in Nature Communications. However, there are a few points that still need to be addressed before publication.

1. The authors add data on measurements of diameter of filaments associated with melanosomes and find that it is consistent with diameter of actin filaments. However, they do not improve quality of EM images, which remains poor and therefore new data are not convincing. The authors should obtain better images of actin filaments preferably decorated with myosin S1 or remove this section from the paper.

To address this concern we have revised figure 3 to include images showing the results of EM imaging of phalloidin immunogold and myosin S1 labelling of filaments that, along with previously presented data on the filament dimensions, confirms that these are composed of actin. These data are presented in a revised figure 3 (panels D-F) and described in the results section (third yellow highlighted section of text on page 8-9 and methods described on pages 32-34).

2. In revised paper, the authors describe in more detail experiments that involves treatment cells with Latrunculin (page 8, and new Fig. 2). They find that disruption of actin filaments with Latrunculin reduces dispersion of melanosomes in control melan-a cells but not in melan-f- or SPIR1/2-depleted melanocytes. The authors conclude that FMN1 (and SPIRE1/2) assemble dynamic actin filaments that support myosin-Va-dependent melanosome dispersion. However, in melan-f- or SPIR1/2-depleted melanocytes melanosomes are already clustered in the cell center and it is not surprising that Latrunculin treatment does not change their distribution. Decrease in degree of melanosome

dispersion in control melan-a cells indeed suggests that distribution of melanosomes depends on dynamics of actin filaments. However, similarities in melanosome distribution (clustering in the cell center) in Latrunculin-treated melan-a and melan-f- or SPIRE1/2-depleted cells does not prove that FMN1 or SPIRE proteins are involved in assembly of dynamic actin filaments.

We agree with the reviewer that we cannot exclude the involvement of other AF regulators in dynamic AF assembly and function in melanocytes. Indeed, throughout the paper we present data showing that the extent of melanosome dispersion in melan-f and SPIRE1/2 depleted cells is greater than for Rab27a deficient cells. This indicates that FMN/SPIRE independent AFs allow a reduced level of melanosome dispersion compared with wild-type cells. Consistent with this we have also found that depletion of Rab27a in melan-f melanocytes further reduces pigment dispersion (data not shown). To reflect this we have revised the closing sentence of the first paragraph on page 8 that reports the effects of latrunculin-A treatment upon melanosome distribution as follows (see below plus the second block of yellow highlighted text on page 8).

'These data indicate that FMN1 (and SPIRE1/2) are important factors for the assembly of dynamic AFs that support myosin-Va-dependent melanosome dispersion.'

Minor points:

Legend to Fig. 1G says that arrows indicate cells with hyperdispersed melanosomes. However, these cells are marked with asterisks.

We have corrected this in the revised the manuscript (see yellow highlighted text on page 44).

Legend to Fig. 2 is confusing. The legend says that "melan-a and melan-f cells were ... transfected with siRNA indicated..." but images presented here do not seem to show transfection of melan-f cells with siRNA. The authors should revise the legend and describe clearly images shown in Fig. 2.

We have revised the Fig. 2 and the accompanying legend to incorporate new data, as requested by reviewer 2. We hope that the reviewer finds that this revision clarifies the description of the data presented in Fig. 2 and addresses the previous confusion adequately (see yellow highlighted text on page 45).

Reviewers' comments:

REVIEWERS' COMMENTS

Reviewer #1 (Remarks to the Author):

In the revised manuscript, the authors properly addressed all the concerns raised by this reviewer. Thus, I now recommended it for publication in Nature Communications.

Reviewer #2 (Remarks to the Author):

All of my concerns have been fully addressed, and enthusiastically support publication of this manuscript.

Reviewer #3 (Remarks to the Author):

The authors have addressed all my comments. I recommend the paper for publication.

We thank the reviewers for their contribution to revision of the manuscript.